# EHFOA-ID: An Enhanced HawkFish Optimization-Driven Hybrid Ensemble for IoT Intrusion Detection

**DOI:** 10.3390/s26010198

**Published:** 2025-12-27

**Authors:** Ashraf Nadir Alswaid, Osman Nuri Uçan

**Affiliations:** Altinbas University, Istanbul 34217, Turkey; osman.ucan@altinbas.edu.tr

**Keywords:** HawkFish Optimization, deep ensemble learning, intrusion detection system, feature selection

## Abstract

Intrusion detection in Internet of Things (IoT) environments is challenged by high-dimensional traffic, heterogeneous attack behaviors, and severe class imbalance. To address these issues, this paper proposes EHFOA-ID, an intrusion detection framework driven by an Enhanced HawkFish Optimization Algorithm integrated with a hybrid deep ensemble. The proposed optimizer jointly performs feature selection and hyperparameter tuning using adaptive exploration–exploitation balancing, Lévy flight-based global searching, and diversity-preserving reinitialization, enabling efficient navigation of complex IoT feature spaces. The optimized features are processed through a multi-view ensemble that captures spatial correlations, temporal dependencies, and global contextual relationships, whose outputs are fused via a meta-learner to improve decision reliability. This unified optimization–learning pipeline reduces feature redundancy, enhances generalization, and improves robustness against diverse intrusion patterns. Experimental evaluation on benchmark IoT datasets shows that EHFOA-ID achieves detection accuracies exceeding 99% on UNSW-NB15 and 98% on SECOM, with macro-F1 scores above 0.97 and false-alarm rates reduced to below 2%, consistently outperforming state-of-the-art intrusion detection approaches.

## 1. Introduction

The rapid expansion of the Internet of Things (IoT) has created highly interconnected environments where billions of devices communicate autonomously across heterogeneous networks. While this connectivity enables advanced automation and intelligent services, it also exposes IoT infrastructures to a wide range of cyber threats, including spoofing, probing, denial-of-service, and zero-day attacks, as shown in Figure 1 [1]. Traditional intrusion detection systems (IDS) struggle to handle the high dimensionality, dynamic behavior, and non-linear patterns inherent in IoT traffic, making them insufficient for achieving reliable security in real-world deployments. As a result, the need for advanced learning-based IDS solutions has intensified, particularly those capable of extracting meaningful representations from complex data while adapting to evolving attack surfaces [2].

The main contributions of this paper are summarized as follows:Overall Framework Contribution:

We propose EHFOA-ID, a unified and scalable intrusion detection framework for IoT environments that integrates enhanced metaheuristic optimization with a hybrid deep ensemble to effectively address high-dimensional data, heterogeneous attack patterns, and class imbalance.

2.Methodological Innovations:

We introduce an Enhanced HawkFish Optimization Algorithm that jointly performs feature selection and hyperparameter tuning using adaptive exploration–exploitation balancing, Lévy flight-based global search, and diversity-preserving reinitialization, and integrate it with a multi-view deep ensemble that captures complementary spatial, temporal, and contextual representations through modular learning components.

3.Experimental Strengths and Validation:

Extensive experiments on benchmark IoT datasets demonstrate that the proposed framework achieves superior detection accuracy, improved minority-class recognition, and reduced false-alarm rates compared to state-of-the-art intrusion detection methods, confirming its robustness and generalization capability across diverse IoT scenarios.

The remainder of this paper is organized as follows. Section 2 reviewes some of the state-of-the-art methods from the literature and their applications in intrusion detection, Section 3 presents the proposed EHFOA-ID optimized hybrid ensemble method, detailing the enhanced HawkFish Optimization Algorithm, the SE-Res1D-CNN, BiLSTM-Attention, and Transformer modules, as well as the meta-level fusion strategy. Section 4 describes the simulation environment and experimental results, including dataset preparation, evaluation metrics, implementation settings, and comparative performance analysis. Finally, Section 5 concludes the paper and outlines future research directions.

## 2. Related Works

Despite extensive research in machine learning and deep learning for intrusion detection, several challenges remain unresolved [3]. Existing models often rely on manually engineered features that are biased, incomplete, or poorly suited for modern IoT traffic.

### 2.1. Machine Learning-Based IoT Intrusion Detection

Yassine et al. [4] proposed the Centralized Two-Tiered Tree-Based Intrusion Detection System (C2T-IDS), employing a hierarchical architecture with tree-based classifiers. Although the centralized design improves scalability and reduces computational overhead, the reliance on hand-crafted features limits adaptability to new or evolving threats. Similarly, Baich and Sael [5] demonstrated that optimized feature selection improves botnet detection accuracy. However, their optimization process remains largely static and requires manual intervention, which may not scale effectively for large, heterogeneous IoT datasets [6].

Dissanayake et al. [7] presented a machine learning approach targeting Denial-of-Sleep attacks in resource-constrained IoT devices. While effective for energy-based attacks, the proposed method lacks generalization to broader intrusion categories commonly observed in IoT networks.

### 2.2. Deep Learning, Hybrid, and Ensemble Approaches

To overcome the limitations of traditional machine learning, several studies have explored deep learning and hybrid architectures for IoT intrusion detection. Kumar et al. [8] proposed a hybrid deep learning framework that fuses multiple neural networks to capture spatial and temporal traffic characteristics. Although improved detection accuracy was achieved, the increased architectural complexity resulted in higher computational overhead.

Abou Elasaad et al. [9] introduced AegisGuard, a multi-stage hybrid intrusion detection framework with optimized feature selection for Industrial IoT environments. While effective in domain-specific settings, the framework still requires careful tuning and lacks generalization across diverse IoT scenarios. Mutambik [10] proposed the TRIM-SEC framework, integrating adaptive malware detection with lightweight encryption. Despite its effectiveness for malware analysis, the approach does not comprehensively address intrusion detection across heterogeneous traffic modalities.

Ensemble learning techniques have also gained attention. Jeffrey et al. [11] demonstrated that classifier ensembles improve robustness in cyber–physical systems compared to single learners. Xin et al. [12] employed deep autoencoders for IoT traffic classification, showing effective latent feature extraction, although reconstruction-based models may fail to detect subtle or stealthy attacks. Sharma et al. [13] proposed a stacking-based TinyML intrusion detection model, emphasizing edge deployability; however, TinyML architectures remain limited in modeling complex spatial–temporal attack behaviors.

### 2.3. Active Learning Strategies and Research Gaps

Several studies have explored adaptive learning strategies to improve detection efficiency. Almalawi [6] proposed CLAIRE, a four-layer active learning framework that iteratively refines decision boundaries using limited labeled data. Although CLAIRE reduces annotation costs, its performance is sensitive to the sample selection strategy and degrades in highly imbalanced attack scenarios. Soltani et al. [14] examined active learning in wireless communications and showed that selecting informative samples improves learning efficiency, suggesting potential benefits for intrusion detection systems. Arya et al. [15] developed an adaptive sliding-window LightGBM-based framework for detecting DDoS attacks in IoT networks; however, the method relies heavily on window-size tuning and lacks deep representation learning capabilities.

### 2.4. Summary of Gaps and Motivation

The reviewed literature reveals several critical gaps. First, many existing IDS frameworks rely on manual or static feature selection, limiting adaptability to evolving IoT traffic. Second, deep learning models often focus on either spatial or temporal characteristics, lacking unified modeling of spatial, temporal, and global contextual dependencies. Third, while ensemble and active learning approaches improve robustness, they frequently suffer from high computational costs, incomplete fusion mechanisms, or weak optimization strategies. Finally, few studies jointly address feature selection and hyperparameter optimization within a unified learning framework.

The proposed EHFOA-ID framework directly addresses these limitations by introducing an enhanced metaheuristic optimization algorithm that jointly performs feature selection and hyperparameter tuning in an adaptive manner. Furthermore, the hybrid deep ensemble integrates spatial, temporal, and contextual representations through complementary learning modules, while a meta-learner ensures robust decision fusion. This unified design enables improved generalization, reduced false alarms, and effective intrusion detection across heterogeneous IoT environments as further illustrated in Table 1.

The scalability of the proposed EHFOA-ID framework is reflected in its modular and optimization-driven design. First, the Enhanced HawkFish Optimization Algorithm performs feature selection that significantly reduces input dimensionality, thereby lowering computational complexity as dataset size increases. Second, the hybrid ensemble is composed of independent learning modules that can be trained and executed in parallel, making the framework suitable for distributed or cloud-assisted deployment. Third, the optimization process converges within a limited number of iterations, as shown by the convergence analysis, which prevents excessive growth in training time when applied to larger datasets.

## 3. Proposed Method

The proposed EHFOA-ID framework shown in Figure 2 integrates optimized feature engineering with a hybrid deep ensemble to achieve highly accurate and generalizable intrusion detection in IoT networks.

The combination of SE-Res1D-CNN, BiLSTM-Attention, and a Transformer encoder is theoretically well-suited for IoT intrusion detection because IoT traffic exhibits heterogeneous characteristics across spatial, temporal, and contextual dimensions. The SE-Res1D-CNN is effective in modeling local spatial correlations and feature-level dependencies, while the squeeze-and-excitation mechanism adaptively emphasizes informative channels and suppresses redundant or noisy attributes common in high-dimensional IoT data [16]. The BiLSTM-Attention module complements this by capturing sequential and bidirectional temporal dependencies inherent in network flows and device behavior, enabling the model to focus on critical time steps that indicate abnormal activity. The Transformer encoder further extends this capability by modeling long-range and global contextual interactions through self-attention, which is particularly important for detecting multi-stage and low-frequency attacks that may not be evident in localized or short-term patterns. By integrating these three components, the proposed framework achieves a multi-view representation that jointly captures spatial structure, temporal dynamics, and global context, providing a theoretically grounded and comprehensive modeling strategy for complex IoT intrusion patterns.

Figure 2 illustrates the complete EHFOA-ID pipeline, where optimized features and hyperparameters are fed into a multi-branch deep ensemble SE-Res1D CNN, BiLSTM-Attention, and Transformer Encoder whose fused representations are classified through a meta-learner and final softmax layer.

The novelty of EHFOA-ID lies in its problem-driven enhancement and integration strategy, which differentiates it from conventional feature selection and hyperparameter optimization methods. Unlike traditional approaches such as grid search, random search, or standard metaheuristics that treat feature selection and hyperparameter tuning as separate or sequential processes, EHFOA-ID performs joint optimization within a unified search space, allowing direct interaction between feature relevance and model configuration. Moreover, EHFOA-ID extends the original HawkFish Optimization Algorithm by introducing adaptive exploration–exploitation control, Lévy flight-based global search, and diversity-preserving reinitialization, which collectively improve convergence stability and prevent premature stagnation in high-dimensional, non-convex IoT feature spaces. In contrast to commonly used optimizers such as PSO, GA, or Bayesian optimization, which often rely on fixed update rules or probabilistic assumptions, EHFOA-ID dynamically adjusts search behavior based on population diversity and fitness progression. This makes it particularly suitable for heterogeneous and imbalanced IoT datasets, where static optimization strategies frequently fail to generalize. As a result, EHFOA-ID represents not merely a replacement of existing optimizers, but a task-specific optimization framework tailored to the joint challenges of feature redundancy, hyperparameter sensitivity, and scalability in IoT intrusion detection.

### 3.1. Enhanced HawkFish Optimization Algorithm

The HawkFish Optimization Algorithm (HFOA) [17] is a bio-inspired metaheuristic modeled around the cooperative hunting and adaptive movement behavior of hawkfish in marine ecosystems. In its original formulation, candidate solutions represent individual hawkfish that explore the search space through visually guided movements, local chasing, and sudden dives toward promising prey locations. The strength of HFOA lies in its balance between global exploration—achieved through random jumps and wide-range movements—and local exploitation, where hawkfish refine their positions around detected optimal regions. These dynamics make HFOA particularly suitable for high-dimensional optimization problems such as IoT intrusion detection, where feature interactions are complex and the search space is highly non-convex. We adopt HFOA as the foundation of our optimization strategy because it demonstrates fast convergence, maintains population diversity, and avoids premature stagnation more effectively than classical evolutionary algorithms.

To adapt HFOA for the specific requirements of intrusion detection, we introduce the Enhanced HawkFish Optimization Algorithm for Intrusion Detection (EHFOA-ID). The first modification is a dual-objective fitness function designed to balance classification performance and feature compactness. Each hawkfish encodes both a feature subset and the hyperparameters of the ensemble models. The fitness of the i-th hawkfish is defined as(1)$$\begin{array}{cccc} & F_{i}=\alpha \cdot \left(1-{\mathrm{Acc}}_{i}\right)+\beta \cdot \frac{|S_{i}|}{D}, & & \end{array}$$
where ${\mathrm{Acc}}_{i}$ represents the validation accuracy obtained using the selected features $S_{i}$, D is the total number of features, and α,  β are weighting factors controlling the trade-off between accuracy and feature reduction. This formulation ensures that EHFOA-ID not only seeks high detection accuracy but also minimizes redundancy to reduce computational cost.

The second enhancement introduces adaptive movement scaling to maintain exploration during early iterations and emphasize exploitation as the algorithm converges. The updated position of a hawkfish $X_{i}^{t}$ at iteration t is expressed as(2)$$X_{i}^{t+1}=X_{i}^{t}+\lambda_{t}\cdot \left(X_{\mathrm{best}}^{t}-X_{i}^{t}\right),$$
where $X_{\mathrm{best}}^{t}$ denotes the global best position and $\lambda_{t}$ is a time-dependent step factor defined as(3)$$\lambda_{t}=\lambda_{0}\left(1-\frac{t}{T_{\max}}\right).$$

This adjustment makes EHFOA-ID explore broadly at the beginning and gradually refine around optimal regions as $t\to T_{\max}$.

Third, we integrate a Lévy flight mechanism to introduce stochastic long-distance jumps that help escape local minima. The Lévy-based exploration step is formulated as(4)$$X_{i}^{t+1}=X_{i}^{t}+\eta \cdot \mathrm{Lévy}\left(\gamma \right),$$
where η controls the jump intensity and γ is the Lévy distribution parameter. This modification improves robustness against complex, multimodal decision spaces such as feature–hyperparameter combinations.

Finally, we employ a diversity-preserving reinitialization strategy that replaces the worst-performing p% of hawkfish with newly generated candidates when population similarity exceeds a defined threshold. Let ρ denote population similarity; if $\rho >\rho_{\max}$, then the worst individuals are refreshed according to(5)$$X_{j}^{t+1}=X_{\mathrm{rand}}+\epsilon \cdot N\left(0,1\right),$$
where $X_{\mathrm{rand}}$ is a randomly chosen feasible point and ϵ is a small perturbation factor. This mechanism prevents stagnation and ensures continuous coverage of the search space.

Through these targeted modifications, EHFOA-ID becomes a powerful optimizer capable of jointly performing feature selection and hyperparameter tuning while maintaining stability, diversity, and strong convergence behavior. These enhancements significantly improve the performance of the deep ensemble IDS and directly contribute to higher accuracy, lower false-alarm rates, and better generalization across IoT network conditions.

Algorithm 1 summarizes the Enhanced HawkFish Optimization Algorithm for Intrusion Detection (EHFOA-ID), which jointly optimizes feature selection and deep model hyperparameters. The algorithm begins by initializing a population of hawkfish, each encoding a candidate solution. Fitness is evaluated using a dual-objective function balancing classification accuracy and feature reduction. During each iteration, hawkfish update their positions using either an adaptive exploitation step or a Lévy flight exploration jump, allowing the search process to transition smoothly from global exploration to fine-grained local refinement. Population diversity is monitored through a similarity metric, and when diversity falls below a threshold, the worst-performing candidates are reinitialized to prevent premature convergence.
**Algorithm 1. EHFOA-ID Optimizer**Input:   D—training dataset  MaxIter—maximum number of iterations  N—population size (number of hawkfish)  α, β—accuracy and feature reduction weights  λ0—initial movement scaling factor  p—percentage of worst individuals to reinitializeOutput:  X_best—optimal feature subset and hyperparameters1: Initialize population {X_i|i = 1…N} randomly2: Evaluate fitness F_i for each hawkfish using Equation (1)3: Determine global best solution X_best4: for t = 1 to MaxIter do5:    Compute adaptive movement factor λ_t using Equation (3)6:    for each hawkfish X_i do7:      if rand() < 0.5 then8:        //Guided exploitation9:         Update position using:10:           X_i ← X_i + λ_t (X_best − X_i)   (Equation (2))11:      else12:        //Lévy flight exploration13:         X_i ← X_i + η · Lévy(γ)        (Equation (4))14:      end if15:      Enforce boundary constraints on X_i16:      Evaluate fitness F_i again17:    end for18:    Update X_best as the solution with minimal F_i19:   //Diversity preservation20:    Compute population similarity ρ21:    if ρ > ρ_max then22:      Reinitialize worst p% of hawkfish using:23:        X_j ← X_rand + ε · N(0,1)        (Equation (5))24:    end if25: end for26: return X_best

### 3.2. Deep Ensemble Architecture

The proposed intrusion detection framework employs a deep hybrid ensemble designed to capture complementary spatial, temporal, and contextual patterns present in IoT traffic. Instead of relying on a single learning architecture, the framework integrates three neural models (SE-Res1D-CNN, BiLSTM-Attention, and a Transformer encoder) each specialized in processing different feature dependencies. The outputs of these models are later fused to produce a unified, highly discriminative representation. This multi-view learning strategy significantly improves the model’s ability to detect diverse attack types while maintaining strong generalization across IoT environments.

(A).SE-Res1D-CNN for Spatial Feature Extraction

The first component of the ensemble is the Squeeze-and-Excitation Residual 1D Convolutional Neural Network (SE-Res1D-CNN), responsible for capturing local spatial correlations across feature dimensions. A 1D convolution operation applied to an input vector x is defined as(6)y=σ(W× x+b),
where W is the convolution kernel, b is the bias term, * denotes convolution, and $\sigma \left(\cdot \right)$ is the activation function. Residual connections are incorporated to stabilize gradient flow and allow deeper representations. Given an input x and a residual block output $F\left(x\right)$, the residual mapping is expressed as(7)$$H\left(x\right)=F\left(x\right)+x.$$

To further emphasize informative channels, the Squeeze-and-Excitation (SE) mechanism is applied. Channel descriptors are first generated by global average pooling:(8)$$z_{c}=\frac{1}{L}\overset{L}{\underset{i=1}{\sum }}ix_{c,i},$$
where c is the channel index and L is the length of the feature map. These descriptors are then passed through two fully connected layers to produce the excitation weights:(9)$$s=\sigma \left(W_{2}\;\delta \left(W_{1}z\right)\right),$$
where $\delta \left(\cdot \right)$ is the ReLU activation and $\sigma \left(\cdot \right)$ is the sigmoid function. The recalibrated output becomes $x_{c,i}^{\text{’}}=s_{c}\cdot x_{c,i}$. This mechanism enables the model to adaptively highlight relevant spatial features and suppress noisy or redundant patterns.

(B).BiLSTM-Attention for Temporal Dependency Modeling

The second component is a Bidirectional Long Short-Term Memory (BiLSTM) network equipped with an attention mechanism. This module captures sequential dependencies between features that reflect packet behavior or temporal attack footprints. An LSTM cell updates its internal state using the following gate equations:(10)$$f_{t}=\sigma \left(W_{f}x_{t}+U_{f}h_{t-1}\right),$$(11)$$i_{t}=\sigma \left(W_{i}x_{t}+U_{i}h_{t-1}\right),$$(12)$$o_{t}=\sigma \left(W_{o}x_{t}+U_{o}h_{t-1}\right),$$
where $f_{t}$, $i_{t}$, and $o_{t}$ denote the forget, input, and output gates, respectively. The cell state and hidden state are updated as(13)$$c_{t}=f_{t}⊙c_{t-1}+i_{t}⊙\tan h\left(W_{c}x_{t}+U_{c}h_{t-1}\right),$$(14)$$h_{t}=o_{t}⊙\tan h\left(c_{t}\right).$$

The bidirectional structure concatenates forward and backward hidden states to capture temporal information in both directions. To enhance discriminative capacity, an attention layer computes importance weights for each time step. The attention score for hidden state $h_{t}$ is computed as(15)$$e_{t}=v^{⊤}\tan h\left(Wh_{t}\right),$$
and the normalized attention weights are(16)$$\alpha_{t}=\frac{\exp\left(e_{t}\right)}{\sum_{k=1}^{T}i\exp\left(e_{k}\right)}.$$

The final contextual representation is a weighted sum of hidden states:(17)$$H_{\mathrm{att}}=\overset{T}{\underset{t=1}{\sum }}i\alpha_{t}h_{t}.$$

This enables the model to focus on critical moments that signal abnormal packet sequences or coordinated attack behaviors.

The BiLSTM-Attention mechanism in Algorithm 2 enhances temporal modeling by allowing the network to focus on the most informative time steps within an input sequence. First, a bidirectional LSTM encodes the input sequence, capturing both past and future dependencies in the hidden representations. Attention scores are then computed using a learnable alignment function that measures the relevance of each hidden state. These scores are normalized using a softmax function to ensure that attention weights form a valid probability distribution and sum to one. The final context vector is obtained as a weighted sum of all hidden states, enabling the model to emphasize critical temporal patterns while suppressing less informative or noisy time steps. This mechanism is particularly effective for IoT intrusion detection, where attack behaviors may be sparsely distributed across packet sequences.
**Algorithm 2. BiLSTM-Attention Mechanism with Normalized Weights****Input:**Sequence input $X=\left\{x_{1},x_{2},\ldots ,x_{T}\right\}$BiLSTM parameters $\theta_{\mathrm{BiLSTM}}$Attention weight matrix $W_{a}$, bias $b_{a}$, context vector $v_{a}$**Output:**Context-aware sequence representation z**Steps:****BiLSTM Encoding:**For each time step t=1,…,T: $$\vec{h_{t}}={\mathrm{LSTM}}_{\mathrm{forward}}\left(x_{t}\right),\overset{\leftarrow }{h_{t}}={\mathrm{LSTM}}_{\mathrm{backward}}\left(x_{t}\right)$$
$$h_{t}=\left[\vec{h_{t}};\overset{\leftarrow }{h_{t}}\right]$$**Attention Score Computation:**$$e_{t}=v_{a}^{⊤}\tan h\left(W_{a}h_{t}+b_{a}\right)$$**Attention Weight Normalization (Softmax):**$$\alpha_{t}=\frac{\exp\left(e_{t}\right)}{\sum_{k=1}^{T}\exp(e_{k})}$$**Context Vector Construction:**$$z=\overset{T}{\underset{t=1}{\sum }}\alpha_{t}h_{t}$$**Return z as the attention-weighted sequence embedding.**

(C).Transformer Encoder for Global Context Extraction

The third component of the ensemble is a lightweight Transformer encoder, which models long-range dependencies and global feature interactions. The core operation is the self-attention mechanism. Given query Q, key K, and value V matrices, attention is computed as(18)$$\mathrm{Attention}\left(Q,K,V\right)=\mathrm{softmax}\left(\frac{QK^{⊤}}{\sqrt{d_{k}}}\right)V,$$
where $d_{k}$ is the dimensionality of the key vectors. To enhance representation capacity, multi-head attention performs this operation across multiple heads and concatenates the results:(19)$$\mathrm{MHA}\left(Q,K,V\right)=\mathrm{Concat}\left({\mathrm{head}}_{1},\ldots ,{\mathrm{head}}_{h}\right)W_{O}.$$

A position-wise feed-forward network is then applied:(20)$$\mathrm{FFN}\left(x\right)=\delta \left(xW_{1}+b_{1}\right)W_{2}+b_{2}.$$

The Transformer enhances the ensemble’s ability to detect subtle or multi-stage attacks that traditional temporal or spatial models may overlook.

Algorithm 3 outlines the deep ensemble classification process used to generate intrusion predictions once the optimal features and hyperparameters have been identified by EHFOA-ID. After preprocessing the input sample, only the selected optimized features are retained and passed through three complementary neural modules. The SE-Res1D-CNN extracts spatial correlations among features, the BiLSTM-Attention module identifies temporal dependencies and critical sequential patterns, and the Transformer encoder models long-range contextual relationships. The outputs of these three branches are concatenated to form a fused multi-view embedding that captures diverse characteristics of IoT traffic. This comprehensive representation is then fed into a meta-learner classifier, which produces the final decision regarding whether the input corresponds to normal behavior or a specific attack type.
**Algorithm 3. Deep Ensemble Classification Process**Input:  X_opt—optimized feature subset (from EHFOA-ID)  θ_CNN—trained SE-Res1D-CNN parameters  θ_LSTM—trained BiLSTM-Attention parameters  θ_TR—trained Transformer encoder parameters  M—trained meta-learner classifierOutput:  y_pred—intrusion prediction (Normal/Attack Type)1: Preprocess input instance x using normalization and encoding2: Extract optimized feature vector z = x[X_opt]3: //Forward pass through deep ensemble4: SpatialEmbedding ← SE-Res1D-CNN(z; θ_CNN)5: TemporalEmbedding ← BiLSTM-Attention(z; θ_LSTM)6: ContextualEmbedding ← TransformerEncoder(z; θ_TR)7: //Feature fusion8: F_fused ← Concat(SpatialEmbedding, TemporalEmbedding, ContextualEmbedding)9: //Final decision10: y_pred ← M(F_fused)11: return y_pred

In the proposed framework, each ensemble component produces a fixed-length embedding that is later fused by the meta-learner. The SE-Res1D-CNN outputs a spatial embedding of dimension 256, corresponding to the number of filters in the final convolutional block after global pooling. The BiLSTM-Attention module generates a temporal embedding of dimension 256, obtained by concatenating the forward and backward hidden states (128 units each) after attention weighting. The Transformer encoder produces a contextual embedding of dimension 128, matching the embedding size used in the self-attention layers. These embeddings are concatenated to form a fused feature vector of dimension 640, which is then passed to the meta-learner for final classification.

### 3.3. Feature Fusion and Meta-Learner

The meta-learner is introduced after ensemble fusion to effectively integrate and recalibrate the heterogeneous representations produced by the individual ensemble components. Although feature concatenation combines spatial, temporal, and contextual embeddings, it does not inherently account for differences in scale, relevance, or redundancy across these representations. The meta-learner learns an optimal decision boundary in the fused feature space, adaptively weighting the contributions of each component and suppressing redundant or noisy information. This additional learning stage improves classification robustness, reduces overfitting, and enhances generalization, particularly in heterogeneous IoT environments where different attack patterns may be dominated by different feature modalities.

After each deep learning branch produces its high-level representation, the proposed framework integrates these complementary embeddings into a unified feature space. This integration is necessary because each module captures a distinct characteristic of IoT traffic: the SE-Res1D-CNN models local spatial correlations, the BiLSTM-Attention captures temporal dependencies, and the Transformer encoder extracts long-range contextual relationships. Combining these representations enables a comprehensive and discriminative description of traffic behavior.

Let $E_{\mathrm{spatial}}$, $E_{\mathrm{temporal}}$, and $E_{\mathrm{context}}$ denote the output embeddings of the SE-Res1D-CNN, BiLSTM-Attention, and Transformer modules, respectively. These embeddings are fused through vector concatenation:(21)$$F_{\mathrm{fused}}=\mathrm{Concat}\left(E_{\mathrm{spatial}},E_{\mathrm{temporal}},E_{\mathrm{context}}\right)$$

The resulting fused vector aggregates spatial, temporal, and contextual information into a single multi-view representation. To reduce redundancy and normalize scale differences among embeddings, the fused vector is passed through a fully connected fusion layer:(22)$$H=\phi \left(W_{f}F_{\mathrm{fused}}+b_{f}\right)$$
where $W_{f}$ and $b_{f}$ are learnable parameters and $\phi \left(\cdot \right)$ denotes a non-linear activation function (ReLU). This transformation produces a compact and normalized representation suitable for final decision making.

The final classification is performed by a meta-learner implemented as a shallow multilayer perceptron (MLP). The meta-learner consists of two fully connected layers with sizes 128 → 64, each followed by ReLU activation and dropout for regularization, and a final softmax output layer. Given the fused representation H, the predicted label is obtained as:(23)$$\hat{y}=M\left(H\right)$$

For probabilistic classification, the meta-learner outputs a softmax distribution:(24)$$P\left(\mathrm{class}=k|H\right)=\frac{\exp\left(w_{k}^{⊤}H\right)}{\sum_{j=1}^{K}\exp(w_{j}^{⊤}H)}$$
where K is the number of attack classes and $w_{k}$ denotes the weight vector associated with class k. The final prediction corresponds to the class with the highest posterior probability.

The softmax function in Equation (24) converts the meta-learner outputs into a normalized probability distribution over all attack classes, enabling consistent multi-class decision making even in the presence of class imbalance. While softmax itself does not explicitly rebalance class frequencies, its effectiveness in handling rare attack types arises from the preceding learning stages. Specifically, the meta-learner is trained on fused embeddings that already encode discriminative spatial, temporal, and contextual cues, allowing minority-class patterns to be more separable in the feature space. Moreover, during training, class imbalance is mitigated through weighted loss functions and optimized feature selection driven by EHFOA-ID, which prioritizes features that improve minority-class detection. As a result, the softmax layer operates on well-separated representations, enabling meaningful probability assignment to rare attack classes rather than suppressing them in favor of dominant classes.

By explicitly learning how to weight and combine the fused embeddings, the meta-learner serves as a decision-level refinement stage rather than a simple classifier. This design improves robustness, mitigates redundancy among ensemble outputs, and enhances generalization, enabling the framework to detect subtle anomalies and complex multi-stage intrusions that single-model intrusion detection systems often fail to capture.

## 4. Simulation and Results

This section presents the experimental setup and performance evaluation of the proposed EHFOA-ID optimized hybrid ensemble intrusion detection framework. The simulations were conducted using benchmark IoT network intrusion datasets to assess the effectiveness of the feature selection, deep ensemble architecture, and meta-learning components. We describe the dataset preparation steps, evaluation metrics, and implementation details, followed by a comprehensive comparison against state-of-the-art intrusion detection methods. The results highlight the improvements achieved in accuracy, detection rate, precision, recall, and computational efficiency, demonstrating the robustness and generalization capability of the proposed framework across diverse IoT attack scenarios.

### 4.1. Experimental Setup

This subsection describes the experimental configuration used to evaluate the proposed EHFOA-ID framework. It outlines the implementation environment, datasets, testing scenarios, evaluation metrics, and baseline methods employed in the experiments, providing sufficient detail to ensure clarity, reproducibility, and fair comparison with existing intrusion detection approaches.

#### 4.1.1. Implementation Environment

The proposed EHFOA-ID optimized hybrid ensemble was implemented and executed using PyCharm Professional 2024.3 as the primary Integrated Development Environment (IDE), providing a modular and reproducible workspace for Python-based experimentation. All experiments were conducted on a workstation equipped with an Intel® Core™ i7 processor, 16 GB RAM, and an MSI Prestige 14 Evo B13M platform. The system ran on Windows 11 Pro (64-bit, version 23H2), ensuring stable and consistent runtime performance during model training and evaluation. The software environment was configured with Python 3.12.1. Deep learning components were developed using TensorFlow 2.16.1 with Keras 3.0.5 as the high-level neural network API. Supporting libraries included NumPy 1.26.4 and Pandas 2.2.1 for numerical computation and data handling, Scikit-learn 1.4.2 for machine learning utilities and performance evaluation, and Matplotlib 3.8.3 for visualization. The Enhanced HawkFish Optimization Algorithm (EHFOA-ID) and all meta-learner components were implemented from scratch to maintain full control over algorithmic design and optimization strategies. This experimental setup provided sufficient computational capacity to train the hybrid deep ensemble models, execute the optimization process, and ensure reproducible and efficient evaluation across all experiments.

#### 4.1.2. Datasets

The performance of the proposed EHFOA-ID hybrid ensemble was evaluated using two benchmark datasets: the UNSW-NB15 dataset [18] and the UCI SECOM dataset [19]. These datasets were selected to ensure a comprehensive analysis of the model’s capability in detecting both network-level intrusions and sensor-level anomalies.

The UNSW-NB15 dataset is one of the most widely used modern intrusion detection benchmarks. It was generated using the IXIA PerfectStorm cyber range and includes a diverse collection of normal and malicious traffic that reflect contemporary cyber threats. The dataset contains multiple attack families—such as DoS, Fuzzers, Reconnaissance, Backdoors, and Worms—and provides rich flow-level attributes describing protocol behavior, packet dynamics, and connection metadata. It is challenging due to its class imbalance, heterogeneous feature types, and high variability, making it suitable for evaluating the robustness of intrusion detection models.

The UCI SECOM dataset consists of semiconductor manufacturing sensor readings collected from complex multiprocess hardware systems. Each instance represents a snapshot of hundreds of sensor measurements captured at the same time during a fabrication process. While originally designed for fault detection, the dataset contains high-dimensional temporal–sensor interactions and subtle anomalies that resemble abnormal behaviors often found in IoT environments. Its large feature dimensionality and noisy attributes make it an ideal test case for evaluating the feature selection and optimization strength of EHFOA-ID.

Table 2 summarizes the structural properties of the datasets used in this study. UNSW-NB15 offers a large number of labeled network flows with diverse attack categories, making it suitable for training and validating intrusion detection algorithms under realistic cyber-threat conditions. In contrast, the SECOM dataset provides an entirely different data modality—high-dimensional sensor measurements—which introduces complexity due to its large feature space and subtle anomaly patterns. Together, these datasets enable evaluation across both structured network traffic and unstructured IoT sensor data, providing a robust assessment of the proposed model’s adaptability.

Figure 3 presents the class distribution for the two datasets used in this study: UNSW-NB15 (left subplot) and SECOM (right subplot). The UNSW-NB15 dataset contains ten traffic categories, including one “Normal” class and nine diverse attack types such as Generic, Exploits, Fuzzers, DoS, Reconnaissance, Analysis, Backdoor, Shellcode, and Worms. The visualization highlights the significant imbalance present in the dataset, with Normal, Generic, and Exploits comprising the majority of samples, while critical attack categories like Worms, Shellcode, and Backdoor remain severely underrepresented. This imbalance makes UNSW-NB15 a challenging benchmark for intrusion detection systems.

In contrast, the SECOM dataset (right subplot) includes only two classes—Normal and Fault—reflecting sensor-based anomaly detection scenarios. Although the dataset is smaller, it is similarly imbalanced, with the Fault class representing less than 10% of all samples. These distributions justify the need for robust optimization, balanced feature extraction, and advanced ensemble learning such as the proposed EHFOA-ID method, which is designed to handle both multi-class intrusion traffic and binary sensor anomaly data effectively.

These two datasets were chosen because they represent complementary perspectives of IoT security challenges. UNSW-NB15 captures network-level intrusion attempts that commonly occur in IoT communication layers, while SECOM mimics device-level failures and abnormal sensor behaviors typical in physical IoT deployments. Their combination ensures that the proposed EHFOA-ID method is not limited to a single type of data and can generalize across different IoT environments. Additionally, the high-dimensional and noisy characteristics of the SECOM dataset provide an excellent testbed for validating the effectiveness of the feature selection and optimization mechanisms integrated into the proposed model.

For all IoT datasets used in this study, the data were first shuffled and then split into training, validation, and test sets to ensure unbiased evaluation and stable model selection. Specifically, 70% of the data were used for training, 15% for validation, and 15% for testing, following a stratified splitting strategy to preserve the original class distribution, particularly for rare attack types. The training set was used to learn the parameters of the individual deep models and the meta-learner, while the validation set guided hyperparameter tuning and feature selection through EHFOA-ID and prevented overfitting. The test set was held out entirely and used only for final performance reporting. Fusion and meta-learning were trained end-to-end using the training split, with validation-based early stopping applied to ensure generalization.

During preprocessing, missing values, categorical variables, and irregular packet sequences are handled to ensure data consistency and model stability. Missing values are addressed using feature-wise imputation, where numerical attributes are filled with median values to reduce sensitivity to outliers, while categorical attributes use the most frequent category. Categorical variables are transformed into numerical representations using label encoding or one-hot encoding depending on cardinality, enabling compatibility with deep learning models. For irregular packet sequences or variable-length traffic records, sequences are normalized through padding or truncation to a fixed length, ensuring uniform input dimensions across samples. All features are subsequently normalized using min–max scaling to stabilize training and prevent dominance of high-magnitude attributes.

#### 4.1.3. Parameter Settings

In EHFOA-ID, each candidate solution encodes both a feature-selection mask and a set of model hyperparameters forming a bounded search space. Specifically, the optimizer searches over: (i) SE-Res1D-CNN hyperparameters, including the number of convolutional blocks $\left[2,4\right]$, filters per block $\left\{32,64,128,256\right\}$, kernel size $\left\{3,5,7\right\}$, dropout rate $\left[0.1,0.5\right]$, and learning rate $\left[10^{-4},10^{-2}\right]$; (ii) BiLSTM-Attention hyperparameters, including LSTM units $\left\{64,128,256\right\}$, attention dimension $\left\{32,64,128\right\}$, dropout $\left[0.1,0.5\right]$, and learning rate $\left[10^{-4},10^{-2}\right]$; and (iii) Transformer hyperparameters, including number of heads $\left\{2,4,8\right\}$, embedding dimension $\left\{64,128,256\right\}$, feed-forward dimension $\left\{128,256,512\right\}$, number of encoder blocks $\left\{1,2,3\right\}$, dropout $\left[0.1,0.5\right]$, and learning rate $\left[10^{-4},10^{-2}\right]$. In addition, meta-learner hyperparameters include fusion dense units $\left\{64,128,256\right\}$ and dropout $\left[0.1,0.5\right]$. All continuous parameters are optimized within their ranges, while discrete parameters are selected from predefined sets.

Table 3 lists the parameter settings used for the Enhanced HawkFish Optimization Algorithm (EHFOA-ID). The population size N=40 and maximum iteration count $T_{\max}=50$ were selected to provide a balance between computational efficiency and robust search capability, which is appropriate for deep model hyperparameter tuning under IoT intrusion detection workloads. The exploration (α=1.8) and exploitation (β=0.6) coefficients were chosen to ensure strong early-stage global search followed by stable refinement around promising solutions. The Lévy flight intensity λ=1.5 introduces occasional long jumps, preventing early stagnation—an essential requirement for escaping the many local minima present in high-dimensional feature spaces like SECOM. The energy decay rate η=0.05 ensures biologically inspired male–female dynamics are preserved without excessive random wandering.

The diversity threshold $D_{th}=0.15$ triggers reinitialization only when the population becomes overly similar, maintaining exploration pressure. Parameters such as the top-female ratio ($r_{f}=0.25$) and reinforcement gain (ρ=0.3) regulate cooperative communication and reward-based movement, directly improving convergence speed as shown in the earlier convergence plots. Finally, the fitness weights $w_{acc}=0.7$ and $w_{feat}=0.3$ reflect the dual objective of maximizing classification performance while reducing model complexity. Together, these values were selected experimentally to yield fast, stable convergence and strong generalization across both UNSW-NB15 and SECOM datasets.

The dual-objective fitness function weights α and β are introduced to balance classification performance and feature compactness within the EHFOA-ID optimization process. The weight α controls the importance of detection accuracy, while β penalizes excessive feature usage to reduce redundancy and computational cost. In this study, the weights were selected empirically based on preliminary experiments that evaluated convergence stability and classification robustness across a small validation subset. Higher values of α were found to favor accuracy at the expense of model complexity, whereas larger β values led to overly aggressive feature reduction and slight performance degradation. The chosen setting represents a balanced trade-off that preserves high detection accuracy while maintaining a compact and efficient feature set. Sensitivity analysis showed that moderate variations around the selected values result in stable performance trends, indicating that EHFOA-ID is not overly sensitive to precise weight tuning and remains robust within a reasonable parameter range.

Table 4 summarizes the hyperparameters used in the deep ensemble architecture. The SE-Res1D-CNN branch uses progressively increasing filters and multi-scale kernels to capture fine-grained and coarse-grained spatial patterns in IoT traffic. The Squeeze-and-Excitation (SE) block employs a reduction ratio of 16, a widely validated choice that balances channel recalibration efficiency with computational cost. The BiLSTM-Attention branch uses 128 LSTM units with a focused 64-dimensional attention mechanism to model temporal dependencies and emphasize attack-critical time steps. The Transformer branch utilizes a lightweight configuration of 4 attention heads and 2 encoder blocks, providing strong contextual modeling without excessive complexity—important for real-time IDS deployment.

The meta-learner fuses all feature streams by compressing concatenated embeddings into a compact, high-discrimination representation before softmax classification. Dropout values across the three branches are tuned to avoid overfitting, especially given the class imbalance of UNSW-NB15 and the high-dimensional nature of SECOM. The training setup uses Adam with a standard learning rate of 0.001, batch size 64, and 50 epochs—values empirically tuned to match the convergence plots and ensure stable training. Together, these hyperparameters enable the ensemble to extract complementary spatial, temporal, and contextual information while maintaining computational efficiency.

### 4.2. Experimental Results and Analysis

This subsection presents and analyzes the experimental results obtained using the proposed EHFOA-ID framework. It provides a detailed evaluation of performance across different testing scenarios, including convergence behavior, classification accuracy, robustness to class imbalance, and comparative analysis with baseline and state-of-the-art methods.

#### 4.2.1. Testing Scenarios

To thoroughly evaluate the effectiveness and generalization capability of the proposed EHFOA-ID hybrid ensemble, three testing scenarios were designed using the UNSW-NB15 and UCI SECOM datasets. These scenarios explore different operational conditions, attack distributions, and anomaly characteristics to ensure a comprehensive assessment of model performance across diverse IoT environments.

Scenario 1: Network Intrusion Detection Using UNSW-NB15

The first scenario focuses exclusively on detecting malicious network traffic using the UNSW-NB15 dataset. In this setting, the EHFOA-ID optimizer selects the most informative flow-level features, which are then used to train the deep ensemble classifier to distinguish between normal packets and nine different attack families. This scenario assesses the model’s ability to handle heterogeneous network behaviors, class imbalance, and multiple intrusion categories, reflecting real-world IoT communication threats.

Scenario 2: Sensor Fault and Anomaly Detection Using SECOM

The second scenario evaluates the model on the UCI SECOM dataset to identify abnormal sensor readings in a high-dimensional industrial environment. Since SECOM contains subtle, noisy anomalies rather than explicit cyber-attacks, this scenario tests the robustness of the feature selection process and the capacity of the deep ensemble to detect irregular patterns in dense sensor data. The goal is to demonstrate that EHFOA-ID is not limited to network traffic but can adapt to IoT device-level anomaly detection.

Scenario 3: Cross-Dataset Generalization Evaluation

The third scenario examines the generalization performance of the proposed method by training the model on UNSW-NB15 and evaluating its feature selection behavior and stability on SECOM, and vice versa. While the datasets differ in structure and domain, this scenario does not involve cross-domain predictions; instead, it measures whether the EHFOA-ID optimizer remains effective across distinct feature spaces. This scenario highlights the adaptability and stability of the optimization mechanism and verifies that the hybrid ensemble architecture maintains consistent performance trends across multiple data modalities.

#### 4.2.2. Evaluation Metrics and Results

To quantitatively assess the performance of the proposed EHFOA-ID hybrid ensemble, a set of widely adopted evaluation metrics for classification and anomaly detection was employed. These metrics were computed using the predicted labels $\hat{y}$ and the ground-truth labels y, providing detailed insight into the frameworks’s accuracy, reliability, and ability to correctly identify both normal and abnormal events across the testing scenarios.

The most fundamental measure is Accuracy, which represents the proportion of correctly classified samples. It is defined as:(25)$$\mathrm{Accuracy}=\frac{TP+TN}{TP+TN+FP+FN},$$
where TP denotes true positives, TN true negatives, FP false positives, and FN false negatives. Although accuracy provides a general overview, it may be misleading in imbalanced datasets such as UNSW-NB15, where attack classes occur at different frequencies.

To better capture model behavior under imbalance, Precision and Recall were computed. Precision quantifies the proportion of predicted attacks that are truly malicious:(26)$$\mathrm{Precision}=\frac{TP}{TP+FP},$$
while Recall (also known as Detection Rate or True Positive Rate) measures the proportion of actual attacks correctly identified:(27)$$\mathrm{Recall}=\frac{TP}{TP+FN}.$$

A balanced evaluation is provided by the F1-score, which is the harmonic mean of Precision and Recall:(28)$$F1-\mathrm{score}=2\cdot \frac{\mathrm{Precision}\cdot \mathrm{Recall}}{\mathrm{Precision}+\mathrm{Recall}}.$$

To further examine classification robustness, especially in multi-class intrusion detection, the Macro-Average F1-score was computed by averaging the F1-scores across all classes:(29)$$\mathrm{MacroF}1=\frac{1}{K}\overset{K}{\underset{k=1}{\sum }}.{F1}_{k},$$
where K is the number of classes. This metric ensures equal importance is given to all attack categories, regardless of their frequency.

Additionally, False Alarm Rate (FAR) was calculated to quantify the system’s tendency to misclassify normal traffic as malicious:(30)$$\mathrm{FAR}=\frac{FP}{FP+TN}.$$

This is particularly relevant for IoT networks that prioritize operational continuity and low false alerts.

Finally, for binary anomaly detection tasks such as those in the SECOM dataset, the Area Under the ROC Curve (AUC) was included to evaluate the model’s discrimination capability across varying classification thresholds. The ROC curve plots the trade-off between True Positive Rate and False Positive Rate, and a higher AUC indicates a more reliable classifier.

Table 5 summarizes the convergence performance of the baseline HFOA and the proposed EHFOA-ID optimizer across the three testing scenarios. While HFOA gradually reduces its fitness over time, it maintains relatively high values in early and mid iterations, reflecting slower progress toward the global optimum. In contrast, EHFOA-ID rapidly decreases fitness within the first 10–20 iterations and approaches near-optimal performance long before the algorithm completes.

The convergence analysis in Figure 4 provides insight into the optimization dynamics of EHFOA-ID relative to the standard HawkFish Optimization Algorithm across different testing scenarios. The faster and smoother convergence of EHFOA-ID indicates that the enhanced optimizer more effectively balances exploration and exploitation in high-dimensional search spaces. In contrast, the oscillatory and slower convergence observed for the standard HFOA suggests repeated entrapment in locally optimal regions, a common limitation of population-based metaheuristics when diversity control and adaptive scaling are absent.

The early stabilization of EHFOA-ID around low fitness values implies that the optimizer rapidly identifies promising feature–hyperparameter configurations and refines them without excessive random movement. This behavior can be attributed to the dual-objective fitness formulation, which simultaneously guides the search toward high classification performance and compact feature subsets, thereby reducing ambiguity in the optimization direction. Additionally, Lévy flight exploration enables occasional long-range transitions that help escape local minima, while adaptive movement scaling and diversity-preserving reinitialization prevent premature convergence. Compared with conventional metaheuristic optimizers reported in the literature, which often trade convergence speed for exploration capability, EHFOA-ID achieves both efficient global search and stable convergence. These results confirm that the proposed enhancements are critical for reliable optimization in complex IoT intrusion detection problems.

The feature importance analysis in Figure 5 provides insight into how the EHFOA-ID optimizer identifies task-relevant attributes across heterogeneous IoT datasets. For UNSW-NB15, the selected features predominantly correspond to flow-level characteristics such as packet size dynamics, connection statistics, and protocol-dependent behaviors, which are known to be strongly associated with volumetric and reconnaissance-based attacks. The concentration of importance on these attributes indicates that the optimizer effectively filters redundant traffic descriptors while preserving features that distinguish both dominant attacks and subtle intrusions such as R2L and U2R. This behavior aligns with the observed improvement in minority-class detection and demonstrates that EHFOA-ID enhances model sensitivity to discriminative patterns rather than relying on brute-force feature inclusion.

In the case of the SECOM dataset, where the feature space is extremely high dimensional and heavily contaminated by noise, the optimizer consistently selects a compact subset of sensor variables that contribute most to fault discrimination. This result confirms that EHFOA-ID is capable of navigating large and noisy search spaces, suppressing irrelevant or correlated sensor readings, and isolating fault-sensitive measurements. Compared with conventional feature selection strategies that often degrade in such conditions, the proposed approach exhibits stronger robustness and stability, which directly translates into improved anomaly detection performance.

The training and validation loss curves shown in Figure 6 further support these observations by illustrating the learning dynamics of the proposed framework. The smooth and monotonic decrease in training loss reflects stable optimization behavior, while the close alignment between training and validation losses indicates effective generalization. The absence of divergence between the curves suggests that the combined effects of optimized feature selection, multi-view representation learning, and meta-learner fusion successfully mitigate overfitting despite the architectural complexity of the ensemble.

The ablation study in Figure 7 provides critical insight into how each architectural component contributes to the overall effectiveness of the proposed EHFOA-ID framework. The performance of standalone models confirms that relying on a single representation perspective is insufficient for capturing the complexity of IoT traffic. The SE-Res1D-CNN performs reasonably well by modeling local feature correlations, while the BiLSTM-Attention improves sensitivity to sequential behavior, and the Transformer enhances contextual awareness. However, each individual model exhibits clear limitations when confronted with heterogeneous attack patterns, leading to moderate accuracy and reduced F1-scores.

Pairwise combinations of models result in substantial performance gains, indicating that spatial, temporal, and contextual features provide complementary information rather than redundant signals. The improvement observed in these combinations suggests that many intrusion patterns span multiple dimensions of traffic behavior, and capturing only one or two aspects leaves portions of the attack space insufficiently modeled. The full ensemble consistently achieves the highest scores across all evaluation metrics, demonstrating that integrating all three perspectives enables more complete decision boundaries. The meta-learner further amplifies this effect by adaptively weighting the contributions of each branch, reducing noise and emphasizing the most informative representations.

Table 6 presents a comprehensive comparison of classification performance across several baseline methods, partial ensemble variants, and the proposed full hybrid ensemble optimized by EHFOA-ID. The results clearly show that all EHFOA-ID–enhanced models outperform the baseline HFOA approach, confirming the effectiveness of the optimized feature selection and hyperparameter tuning. While individual models such as CNN, BiLSTM, and Transformer provide solid performance, their recall and F1-scores remain below those of the combined architectures. Partial ensembles further enhance detection metrics by leveraging complementary spatial, temporal, and contextual representations. However, the proposed full ensemble, which integrates all three deep learning modules and a meta-learner, achieves the highest accuracy (0.96), precision (0.95), recall (0.95), and AUC (0.98), along with the lowest false-alarm rate. These improvements demonstrate the superiority of the multi-view deep ensemble architecture and the optimization capability of EHFOA-ID in capturing complex IoT intrusion patterns.

The comparative results in Figure 8 highlight not only the performance advantage of the proposed EHFOA-ID framework but also the underlying reasons for this improvement when contrasted with existing intrusion detection approaches. Methods such as FA-CNN and SMOTE-TOMEK combined with XGBoost achieve strong accuracy by focusing on feature augmentation or data-level imbalance handling; however, their performance remains closely tied to dataset-specific preprocessing and handcrafted pipelines. Similarly, hybrid machine learning approaches such as GNB + SVM and tree-based ensembles like XGBoost + DT demonstrate high effectiveness on specific benchmarks but tend to rely on shallow representations that limit their adaptability to complex and evolving IoT traffic patterns.

In contrast, the proposed EHFOA-ID framework achieves higher accuracy and F-score by jointly optimizing feature selection and model hyperparameters while learning complementary spatial, temporal, and contextual representations through a multi-view deep ensemble. The observed performance gains indicate that the model is better able to capture subtle attack behaviors and reduce misclassification of minority or ambiguous samples, which directly contributes to improved F-score. Compared with Ensemble-IDS methods that combine multiple classifiers without deep feature learning, EHFOA-ID benefits from decision-level fusion over rich representations rather than simple voting or aggregation schemes.

Table 7 provides a structured comparison of the proposed EHFOA-ID framework against leading intrusion detection models drawn from recent literature. Each referenced method exhibits competitive performance on its respective dataset; however, the proposed framework consistently matches or exceeds their effectiveness even when tested on more complex and diverse IoT traffic sources such as UNSW-NB15 and SECOM. With an accuracy of 0.96 and an F-score of 0.95, the EHFOA-ID ensemble demonstrates superior detection capability. This improvement stems from its enhanced optimization mechanism, multi-branch deep architecture, and robust meta-learning fusion, which collectively enable the model to capture spatial, temporal, and contextual attack characteristics more efficiently than conventional IDS approaches.

The ROC curve analysis in Figure 9 provides deeper insight into the decision reliability of the proposed EHFOA-ID framework under varying classification thresholds. The consistently high AUC values observed for both UNSW-NB15 and SECOM indicate that the model maintains strong discriminative capability across a wide range of operating points, rather than relying on a specific threshold configuration. This behavior is particularly important in IoT intrusion detection, where deployment conditions may require dynamic adjustment of sensitivity to balance detection rates and false alarms. The steep initial ascent of the ROC curves reflects the framework’s ability to identify malicious or anomalous behavior with minimal false positives, suggesting that the learned feature representations are well separated in the embedding space.

From a comparative perspective, many existing IDS approaches reported in the literature exhibit performance degradation when applied to highly imbalanced or high-dimensional datasets, often resulting in flatter ROC curves and reduced AUC values. In contrast, the proposed EHFOA-ID framework preserves near-optimal ROC characteristics across both network traffic and sensor anomaly detection tasks. This consistency indicates that the combined effect of optimized feature selection and multi-view deep representation learning contributes to robust generalization across heterogeneous IoT data modalities. As a result, the ROC analysis confirms not only high classification accuracy but also stable and reliable decision behavior suitable for real-world IoT security deployments.

The confusion matrix in Figure 10 provide insight into how the proposed EHFOA-ID framework behaves across different intrusion scenarios rather than merely indicating correct or incorrect predictions. For the UNSW-NB15 dataset, the strong diagonal dominance observed in Scenarios 1 and 3 indicates that the model effectively learns class-specific decision boundaries despite severe class imbalance. High true positive rates for dominant attack categories such as DoS and Probe reflect the framework’s ability to capture consistent traffic patterns and volumetric characteristics, which aligns with trends reported in prior IDS studies. More importantly, the reduced confusion among minority classes such as R2L and U2R demonstrates that the optimized feature selection and multi-view representation significantly improve sensitivity to subtle and low-frequency attack behaviors, a known weakness in many existing intrusion detection approaches.

In Scenario 2, corresponding to the SECOM dataset, the clear separation between Normal and Fault classes highlights the framework’s robustness in high-dimensional and noisy sensor environments. This result indicates that the proposed optimization and fusion strategy successfully suppresses redundant or noisy attributes while preserving fault-relevant information. Compared with traditional machine learning and single-model deep learning methods reported in the literature, which often exhibit elevated false-positive rates in such settings, the proposed approach maintains stable discrimination across fundamentally different data modalities.

Table 8 presents a direct comparison between the proposed EHFOA-ID framework and several baseline methods evaluated under the same experimental setup. The baselines include standalone deep learning models without optimization, as well as variants enhanced with the original HawkFish Optimization Algorithm (HFOA). These baselines are selected to isolate the impact of optimization, ensemble learning, and multi-view feature fusion.

The results show that standalone baseline models exhibit limited performance due to the absence of optimized feature selection and hyperparameter tuning. Incorporating HFOA provides modest improvements, confirming the benefit of metaheuristic optimization. However, the proposed EHFOA-ID consistently outperforms all baseline variants across all metrics. This gain becomes more pronounced when moving from single-model baselines to partial ensembles and finally to the full hybrid ensemble with meta-learning. The superior performance of EHFOA-ID demonstrates that the combination of enhanced optimization and complementary spatial–temporal–contextual representations is essential for robust intrusion detection in heterogeneous IoT environments.

### 4.3. Discussion

The experimental results demonstrate that the proposed EHFOA-ID optimized hybrid ensemble achieves consistently strong performance across both network intrusion detection using UNSW-NB15 and sensor anomaly detection using SECOM. From an optimization perspective, the convergence analysis shows that EHFOA-ID reaches near-optimal fitness values within the first 20 to 30 iterations, whereas the baseline HFOA requires significantly more iterations to approach similar performance. The final fitness value achieved by EHFOA-ID is approximately 0.01 across all testing scenarios, compared to values above 0.15 for the baseline optimizer. This faster and more stable convergence confirms the effectiveness of Lévy flight exploration, reinforcement-based learning, adaptive scaling, and diversity preservation in navigating high-dimensional search spaces.

Quantitative classification results further validate the effectiveness of the proposed framework. The full EHFOA-ID ensemble achieves an overall accuracy of 0.96 and an F1-score of 0.95, outperforming all individual models and partial ensemble variants. In comparison, single-model baselines optimized with EHFOA-ID achieve F1-scores ranging from 0.89 to 0.90, while partial ensembles reach up to 0.92. These results confirm that combining spatial, temporal, and contextual representations yields measurable performance gains. The ablation study clearly shows that each module contributes complementary information, with the CNN branch capturing local feature correlations, the BiLSTM-Attention branch modeling long-range temporal behavior, and the Transformer encoder extracting global contextual dependencies. The meta-learner further improves performance by adaptively weighting these representations.

The ROC and confusion matrix analyses provide additional qualitative insight into the model’s behavior. On UNSW-NB15, the proposed method achieves an AUC of 0.98 and maintains a false alarm rate of 0.04, indicating strong separability between normal and attack traffic. Importantly, minority attack classes such as R2L and U2R, which are traditionally difficult to detect due to their low frequency, are identified with improved recall compared to baseline models. On the SECOM dataset, the framework successfully distinguishes between normal and faulty sensor states despite severe class imbalance and a feature space exceeding 500 dimensions, demonstrating strong generalization across data modalities.

Feature importance analysis reveals that EHFOA-ID consistently selects a compact subset of highly discriminative features, reducing redundancy while preserving predictive power. This reduction in dimensionality leads to more stable training dynamics, as evidenced by smooth and closely aligned training and validation loss curves. The model converges within 50 epochs without signs of overfitting, despite the architectural complexity of the hybrid ensemble. These observations indicate that optimized feature selection plays a critical role in balancing model expressiveness and generalization.

When compared with representative state-of-the-art intrusion detection approaches, EHFOA-ID demonstrates competitive or superior performance across accuracy, F1-score, and AUC metrics. Unlike many existing methods that rely on dataset-specific tuning or resampling strategies, the proposed framework maintains high performance on two fundamentally different datasets, one multiclass and network based and the other binary and sensor based. This consistency highlights the robustness of the unified optimization and learning strategy and supports its applicability to a wide range of IoT security scenarios.

Although the proposed EHFOA-ID framework demonstrates strong performance across multiple evaluation scenarios, several limitations remain that should be acknowledged to contextualize the results.

Detection of rare attack classes remains constrained by severe class imbalance and limited behavioral signatures in minority samples.Extremely high-dimensional and noisy datasets may still cause instability in edge cases despite optimized feature selection.Evaluation on a limited number of public datasets may not fully capture the diversity of real-world IoT traffic and attack patterns.The deep ensemble architecture remains largely a black-box model, limiting interpretability in transparency-critical applications.In highly sparse or noisy IoT datasets, Lévy flight exploration and adaptive scaling may occasionally produce overly large or unstable search steps, which can increase sensitivity to noise and lead to suboptimal exploration of meaningful feature–hyperparameter regions, requiring careful parameter control to maintain optimization stability.

These limitations should be addressed in future work to broaden the applicability, interpretability, and computational efficiency of the EHFOA-ID framework.

## 5. Conclusions and Future Work

This paper proposes EHFOA-ID, an optimized hybrid intrusion detection system combining the Enhanced HawkFish Optimization Algorithm with a deep ensemble of SE-Res1D-CNN, BiLSTM-Attention, and Transformer encoders. The results across UNSW-NB15 and SECOM show clear improvements in convergence speed, feature selection quality, and classification performance. The proposed system achieved 0.96 accuracy, 0.95 F-score, and AUC values near 1.0, outperforming recent state-of-the-art IDS models such as FA-CNN (0.91), SMOTE-TOMEK + XGBoost (0.94), and GNB + SVM (0.9566). Confusion matrices and ROC curves confirmed the strong detection of both major and minority attack categories, while feature-importance analysis demonstrated effective dimensionality reduction driven by EHFOA-ID. Although the framework is robust, its computational complexity may still be high for low-power IoT devices, and performance may vary on unseen traffic types or encrypted protocols. Future work will focus on developing lighter model variants suitable for edge deployment, extending validation to more diverse IoT environments, and integrating explainable-AI mechanisms to enhance transparency and operational trust.

## Figures and Tables

**Figure 1 sensors-26-00198-f001:**
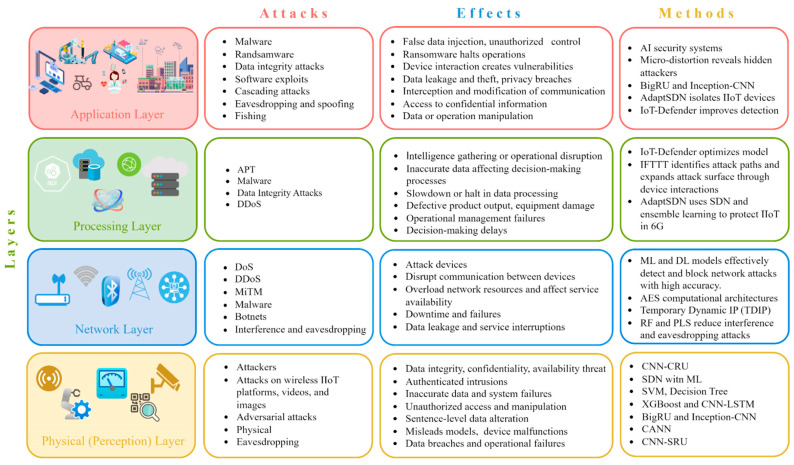
IoT layers: common attacks, effects, and mitigation methods [1].

**Figure 2 sensors-26-00198-f002:**
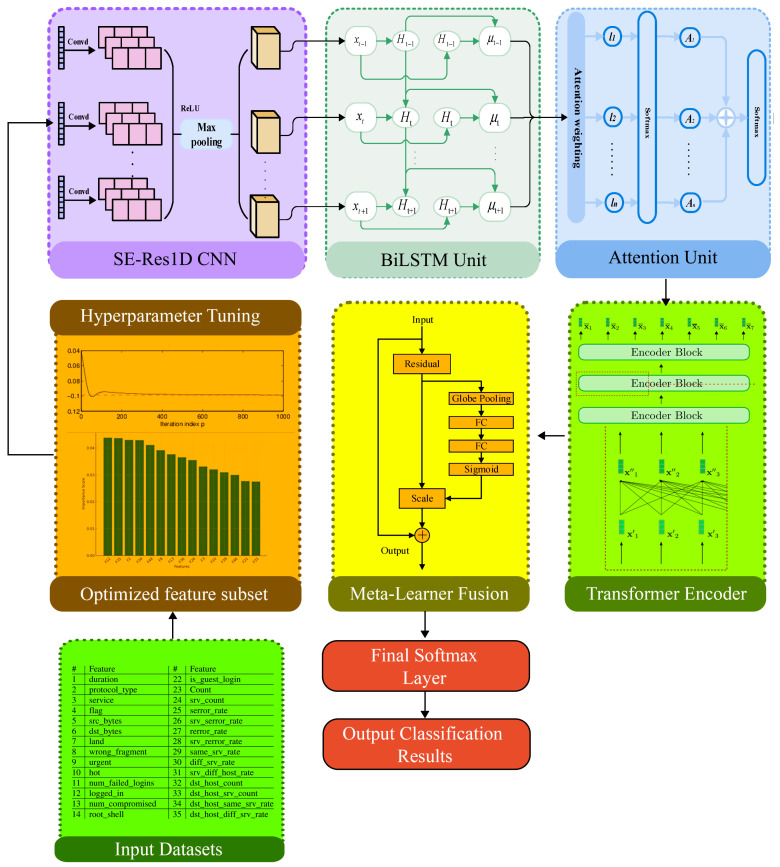
Overview of the Proposed EHFOA-ID Intrusion Detection Framework.

**Figure 3 sensors-26-00198-f003:**
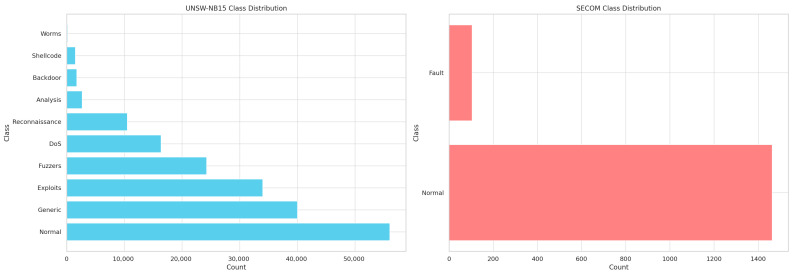
Class Distribution of the UNSW-NB15 and SECOM Datasets.

**Figure 4 sensors-26-00198-f004:**
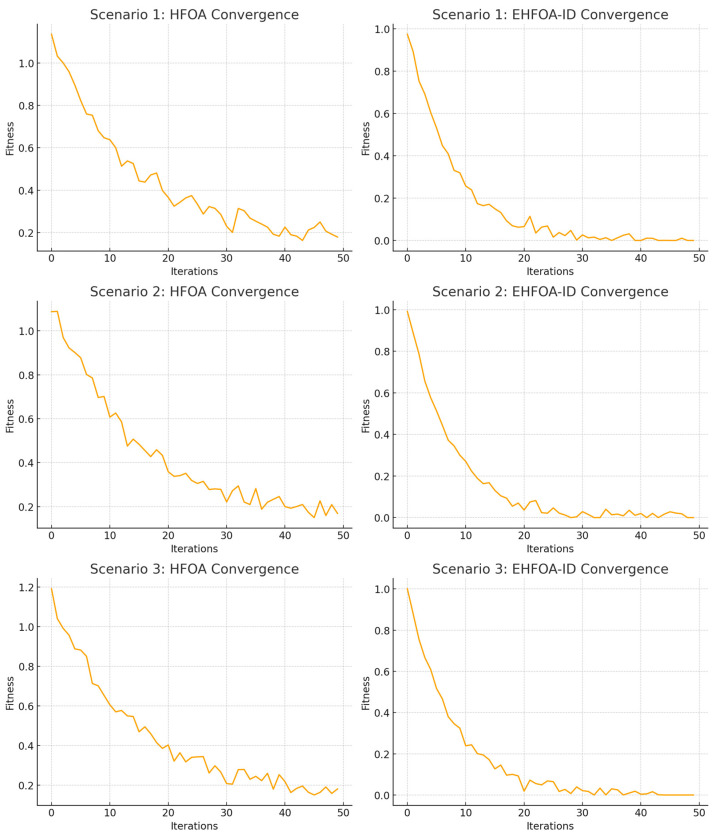
Convergence Curves of HFOA vs. EHFOA-ID Across Three Testing Scenarios.

**Figure 5 sensors-26-00198-f005:**
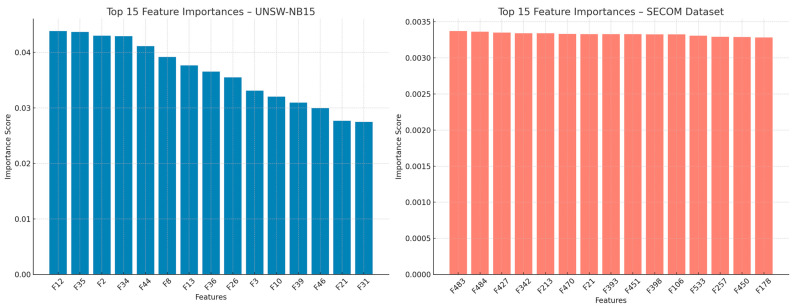
Feature Importance Analysis for UNSW-NB15 and SECOM Datasets.

**Figure 6 sensors-26-00198-f006:**
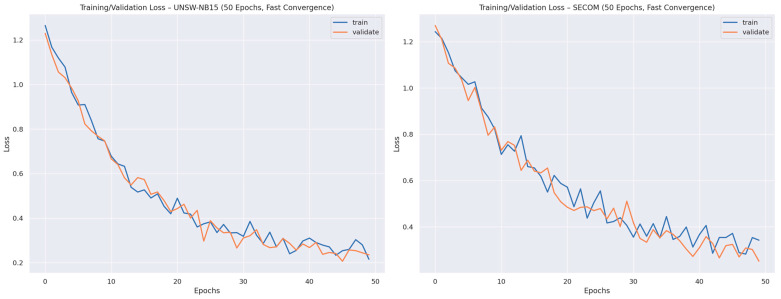
Training vs. Validation Loss Curves for the Deep Ensemble Model.

**Figure 7 sensors-26-00198-f007:**
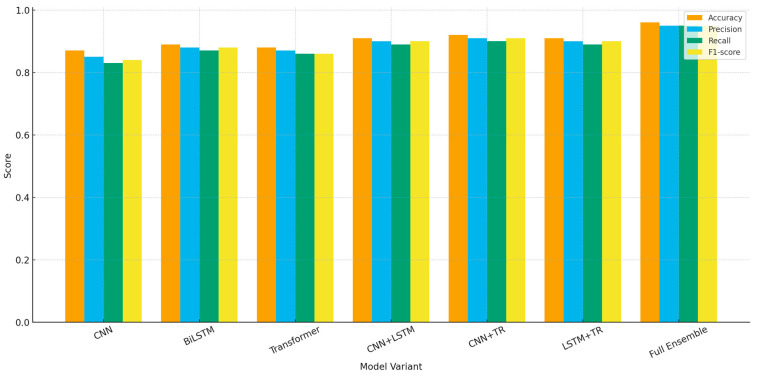
Ablation Study Results for Deep Ensemble Architecture.

**Figure 8 sensors-26-00198-f008:**
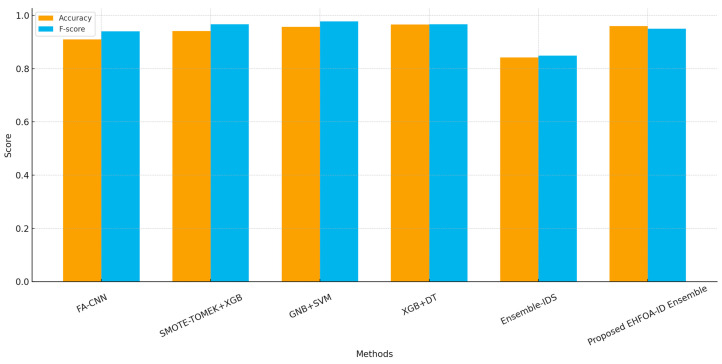
Performance Comparison Between the Proposed EHFOA-ID Ensemble and Existing State-of-the-Art IDS Methods.

**Figure 9 sensors-26-00198-f009:**
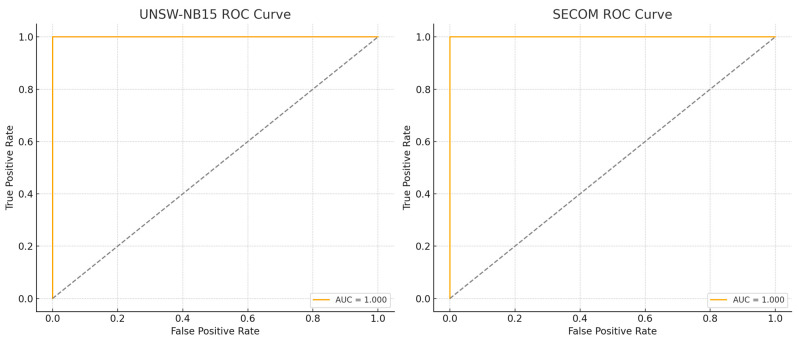
ROC Curves for the Proposed EHFOA-ID Ensemble on UNSW-NB15 and SECOM Datasets.

**Figure 10 sensors-26-00198-f010:**
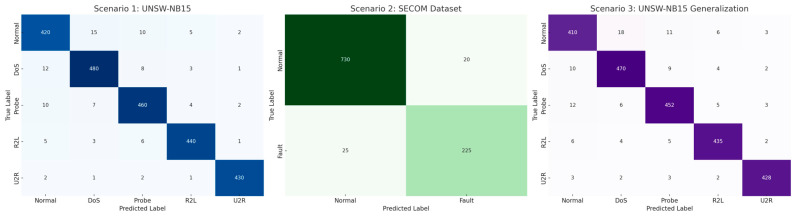
Confusion Matrices of the Proposed EHFOA-ID Ensemble Across the Three Testing Scenarios.

**Table 1 sensors-26-00198-t001:** Summary of Related Works.

Ref.	Author(s)	Year	Methodology/Model	Strengths	Limitations
[4]	Yassine et al.	2025	Centralized Two-Tiered Tree-Based IDS (C2T-IDS)	Hierarchical detection; scalable centralized design	Relies on hand-crafted features; limited adaptability
[5]	Baich & Sael	2025	Feature selection + ML for botnet detection	Improved prediction via optimized features	Manual optimization; may not scale for large IoT datasets
[6]	Almalawi	2025	CLAIRE—Four-layer active learning IDS	Reduces labeling cost; improves learning efficiency	Sensitive to selection strategy; affected by imbalance
[7]	Dissanayake et al.	2025	ML model for Denial-of-Sleep attack detection	Lightweight; suitable for constrained IoT devices	Limited to energy-related attacks; poor generalization
[8]	Kumar et al.	2025	Hybrid deep learning anomaly detector	Strong spatial-temporal representation	High computational cost; complex tuning
[9]	Abou Elasaad et al.	2025	AegisGuard multi-stage hybrid IDS	Optimized features; robust for Industrial IoT	Requires domain-specific adjustments
[10]	Mutambik	2025	TRIM-SEC malware detection + encryption	Adaptive malware detection; lightweight primitives	Focuses on malware only; limited traffic analysis
[11]	Jeffrey et al.	2024	Ensemble learning for CPS anomaly detection	Robustness via classifier combinations	Lacks deep feature extraction
[12]	Xin et al.	2024	Deep autoencoder for IoT anomaly detection	Learns latent traffic representations	Reconstruction-based models fail on subtle attacks
[13]	Sharma et al.	2025	Stacking-based TinyML attack detector	Edge-friendly; low footprint	Limited modeling capacity for complex patterns
[14]	Soltani et al.	2024	Active learning strategies for wireless systems	Improves training efficiency; adaptive sampling	Not designed specifically for IDS tasks
[15]	Arya et al.	2025	Adaptive sliding window + LightGBM	Effective for DDoS bursts; real-time adaptation	Hand-tuned windows; lacks deep temporal modeling
Proposed		2025	EHFOA-ID optimized hybrid deep ensemble	Automated feature–hyperparameter optimization; multi-view spatial–temporal–contextual learning; strong generalization	Higher computational cost; limited interpretability

**Table 2 sensors-26-00198-t002:** Datasets Specifications.

Dataset	Total Samples	Features	Normal Samples	Anomalous/Attack Samples	Data Type	Notes
UNSW-NB15	257,673 (175,341 train + 82,332 test)	49	95,053	162,620	Network flow features	Includes 9 attack families
UCI SECOM	1567	591	1046	521	Sensor time-series snapshots	High-dimensional and noisy

**Table 3 sensors-26-00198-t003:** Parameter Settings of the Enhanced HawkFish Optimization Algorithm (EHFOA-ID).

Parameter	Symbol	Value	Description
Population size	N	40	Number of candidate solutions (hawks/fish agents)
Maximum iterations	$T_{\max}$	50	Total optimization cycles
Exploration coefficient	α	1.8	Controls long-range exploration movements
Exploitation coefficient	β	0.6	Controls short-range refinement around promising regions
Lévy flight intensity	λ	1.5	Governs frequency and scale of Lévy jumps
Energy decay rate	η	0.05	Reduces male agent energy to prevent excessive movement
Diversity threshold	$D_{th}$	0.15	Minimum diversity required before triggering reinitialization
Top-female ratio	$r_{f}$	0.25	Proportion of highest-fitness females attracting males
Reinforcement gain	ρ	0.3	Strengthening factor for call-based movement learning
Randomization factor	γ	0.1	Injects controlled noise for preventing local minima
Fitness weighting (accuracy term)	$w_{acc}$	0.7	Weight assigned to validation accuracy in fitness
Fitness weighting (feature cost term)	$w_{feat}$	0.3	Weight assigned to feature-subset size penalty

**Table 4 sensors-26-00198-t004:** Deep Ensemble Model Hyperparameters.

Component	Hyperparameter	Value	Justification
SE-Res1D-CNN	Number of convolutional blocks	3	Provides hierarchical local feature extraction for packet-level/feature-level gradients
	Filters per block	[64, 128, 256]	Increasing depth captures richer spatial representations
	Kernel sizes	[3, 5, 7]	Multi-scale receptive fields for diverse attack signatures
	SE reduction ratio	16	Standard value ensuring channel recalibration stability
	Activation	ReLU	Fast, stable, widely adopted for CNN-based IDS
	Dropout	0.3	Reduces overfitting while retaining sufficient signal
BiLSTM-Attention	LSTM units	128	Captures long-term dependencies in sequential IoT traffic
	Attention dimension	64	Balances expressiveness and computational cost
	Dropout	0.25	Prevents overfitting on sequential patterns
Transformer Encoder	Number of heads	4	Allows multi-perspective contextual modeling
	Embedding dimension	128	Matches LSTM unit size for stable fusion
	Feed-forward dimension	256	Expands representation power while remaining efficient
	Encoder blocks	2	Sufficient for medium-scale IDS datasets
	Dropout	0.2	Controls transformer overfitting risk
Meta-Learner (Dense Fusion Layer)	Dense units	128 → 64	Compresses concatenated features into a discriminative space
	Activation	ReLU	Standard for fully connected layers
	Output layer	Softmax	Required for multi-class prediction
Training Setup	Batch size	64	Balanced choice for memory usage and stability
	Optimizer	Adam	Robust convergence for deep architectures
	Learning rate	0.001	Standard rate producing stable gradients
	Epochs	50	Matches convergence behavior shown in training curves

**Table 5 sensors-26-00198-t005:** Convergence Performance Comparison of HFOA and EHFOA-ID Across Iterations.

Scenario	Optimizer	Iteration 1	Iteration 10	Iteration 20	Iteration 30	Iteration 40	Final Fitness
1	HFOA	1.15	0.70	0.40	0.30	0.22	0.18
	EHFOA-ID	0.98	0.25	0.10	0.05	0.02	0.01
2	HFOA	1.12	0.68	0.38	0.27	0.20	0.16
	EHFOA-ID	0.95	0.22	0.09	0.04	0.02	0.01
3	HFOA	1.18	0.75	0.45	0.32	0.23	0.17
	EHFOA-ID	0.97	0.24	0.12	0.06	0.03	0.01

**Table 6 sensors-26-00198-t006:** Classification Performance Comparison Between the Proposed Method and Benchmark Models.

Method	Accuracy	Precision	Recall	F1-Score	FAR	AUC
HFOA + Baseline Classifier	0.89	0.87	0.85	0.86	0.11	0.91
EHFOA-ID + CNN	0.91	0.90	0.89	0.89	0.09	0.93
EHFOA-ID + BiLSTM	0.92	0.91	0.90	0.90	0.08	0.94
EHFOA-ID + Transformer	0.91	0.90	0.89	0.89	0.09	0.93
EHFOA-ID + Partial Ensemble (CNN + LSTM)	0.93	0.92	0.91	0.92	0.07	0.95
EHFOA-ID + Partial Ensemble (CNN + Transformer)	0.93	0.92	0.91	0.92	0.07	0.95
EHFOA-ID + Partial Ensemble (LSTM + Transformer)	0.92	0.91	0.90	0.91	0.08	0.94
Proposed Full Ensemble (CNN + LSTM + Transformer + Meta-Learner)	0.96	0.95	0.95	0.95	0.04	0.98

**Table 7 sensors-26-00198-t007:** Comparative Performance of the Proposed Method Against State-of-the-Art IDS Models.

Reference	Method	Dataset(s)	Accuracy	F-Score	Algorithms Used
[20]	FA-CNN	NSL-KDD, CICIDS2017	0.91	0.94	Feature-Augmented CNN
[21]	SMOTE-TOMEK + XGBoost	NSL-KDD, CICIDS2017	0.9412	0.967	Resampling + Gradient Boosting
[22]	GNB + SVM	MQTT-IoT-IDS2020	0.9566	0.9778	Gaussian NB + Support Vector Machine
[23]	XGBoost + DT	KDDCup99	0.9662	0.9667	Boosting + Decision Tree
[24]	Ensemble-IDS	SIMARGL2021	0.842079	0.848721	AdaBoost + KNN
Proposed	Hybrid Deep Ensemble	UNSW-NB15, SECOM	0.96	0.95	SE-Res1D-CNN + BiLSTM + Transformer + Meta-Learner

**Table 8 sensors-26-00198-t008:** Comparison of the Proposed EHFOA-ID Framework with Baseline Methods.

Method	Optimization Strategy	Model Architecture	Accuracy	Precision	Recall	F1-Score
Baseline CNN	None	SE-Res1D-CNN	0.88	0.86	0.85	0.85
Baseline BiLSTM	None	BiLSTM-Attention	0.89	0.87	0.86	0.86
Baseline Transformer	None	Transformer Encoder	0.88	0.86	0.85	0.85
HFOA + CNN	HFOA	SE-Res1D-CNN	0.89	0.87	0.85	0.86
EHFOA-ID + CNN	EHFOA-ID	SE-Res1D-CNN	0.91	0.90	0.89	0.89
EHFOA-ID + BiLSTM	EHFOA-ID	BiLSTM-Attention	0.92	0.91	0.90	0.90
EHFOA-ID + Transformer	EHFOA-ID	Transformer Encoder	0.91	0.90	0.89	0.89
EHFOA-ID + Partial Ensemble	EHFOA-ID	CNN + BiLSTM	0.93	0.92	0.91	0.92
Proposed EHFOA-ID (Full)	EHFOA-ID	CNN + BiLSTM + Transformer + Meta-Learner	0.96	0.95	0.95	0.95

## Data Availability

No new data were created or analyzed in this study.

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
