# Peer review of "EHFOA-ID: An Enhanced HawkFish Optimization-Driven Hybrid Ensemble for IoT Intrusion Detection"

_sensors, 2025, doi:10.3390/s26010198_

Round 1
Reviewer 1 Report
Comments and Suggestions for Authors
- The title contains too many technical terms. Some of them (e.g., BiLSTM-Attention and Transformer) are fairly common, so they are not suitable to highlight as key technical points in the title. I suggest shortening the title and emphasizing one main technique (e.g., the Enhanced HawkFish Optimization Algorithm). You may follow a structure like:
Method name: A hybrid intrusion detection framework based on [a specific technique].
- The abstract includes too much description of the experimental results, and it repeats a lot of content from the conclusion section. I suggest reducing the amount of experimental results in the abstract and adding more details about the proposed method itself. In the abstract, one or two sentences about results are enough.
- There are too many keywords. Please limit them to 3–5. Common techniques do not need to be included as keywords.
- I suggest replacing the word “system” with “framework,” because “system” sounds too engineering-oriented and less academic.
- In Figure 1, “processing layer” is misspelled. Please also check for other spelling mistakes throughout the paper.
- The manuscript mixes the Introduction and Related Work together. This makes the Introduction too long and poorly structured. For journal papers, I suggest writing Introduction and Related Work as separate sections.
- The Related Work section only lists previous studies and does not categorize them. I suggest the authors reorganize and describe prior work using their own logic. I recommend using one or more taxonomies in the revision to group existing methods, and clearly explain the pros and cons of each group. Make Related Work an independent section, and at the end of that section, explain what weaknesses of existing methods are addressed by the proposed approach. Please take the writing of Related Work seriously.
- If the limitations of the current method include many items, they can be listed using numbered points.
- Line 110 says the proposed method is scalable. Where exactly is the scalability reflected? This does not seem to be explained in that paragraph.
- When presenting the contributions in the Introduction, it is strongly recommended to list them in numbered points, and usually condense them into three contributions. The first contribution should describe the overall contribution. The second should focus on methodological innovations (i.e., what is novel in each sub-method). The third should highlight the key strengths in experiments (i.e., the last contribution in the original manuscript).
- It is suggested to delete Figure 2. The workflow diagram is too engineering-oriented, and the figure quality is not very good. Instead, place Figure 3 here. Figure 3 is more academic and is suitable for giving readers a more intuitive understanding of the proposed method before introducing detailed components.
- The subsections in the Experiments section should be reorganized into three subsections. The first subsection should be Experimental Setup, including: Implementation Environment, Datasets, Testing Scenarios, Evaluation Metrics, and Baselines. You can split these parts using third-level headings. The second subsection should be Experimental Results and Analysis, also divided into third-level headings rather than putting all results together. The third subsection should be Discussion.
- The Experiments section does not introduce the baselines. This must be added in the revision.
Overall, the proposed method shows a certain level of innovation, and the experimental results are quite comprehensive. However, the authors still need to further revise the paper to improve its overall logic.
Author Response
REVIEWER 1
The title contains too many technical terms. Some of them (e.g., BiLSTM-Attention and Transformer) are fairly common, so they are not suitable to highlight as key technical points in the title. I suggest shortening the title and emphasizing one main technique (e.g., the Enhanced HawkFish Optimization Algorithm).
Thank you for this constructive suggestion. We agree that the original title was overly detailed and placed unnecessary emphasis on commonly used deep learning components. Following the reviewer’s recommendation, we have revised the title to be more concise and to highlight the primary methodological contribution, namely the Enhanced HawkFish Optimization Algorithm.
The new title is:
“EHFOA-ID: An Enhanced HawkFish Optimization–Driven Hybrid Ensemble for IoT Intrusion Detection”
This revised title better reflects the core novelty of the proposed framework, while still conveying the use of a hybrid ensemble approach without explicitly listing standard architectures. The manuscript text already provides detailed descriptions of the ensemble components, making their inclusion in the title unnecessary. We believe this change improves clarity, focus, and alignment with the journal’s expectations.
The abstract includes too much description of the experimental results, and it repeats a lot of content from the conclusion section. I suggest reducing the amount of experimental results in the abstract and adding more details about the proposed method itself. In the abstract, one or two sentences about results are enough.
We thank the reviewer for this insightful comment. We agree that the original abstract placed excessive emphasis on experimental results and repeated information already presented in the conclusion. In response, we have substantially revised the abstract to focus primarily on the methodological contributions and design of the proposed EHFOA-ID framework.
Specifically, the revised abstract now:
- Emphasizes the Enhanced HawkFish Optimization Algorithm, detailing its role in joint feature selection and hyperparameter tuning.
- Clearly describes the integration of the optimizer with a hybrid deep ensemble and meta-learner fusion strategy.
- Reduces experimental reporting to one concise sentence, summarizing overall performance without repeating detailed metrics.
We believe these changes improve clarity, avoid redundancy with the conclusion, and align the abstract more closely with the reviewer’s recommendation and the journal’s style guidelines.
There are too many keywords. Please limit them to 3–5. Common techniques do not need to be included as keywords.
We appreciate the reviewer’s suggestion. In accordance with this comment, we have reduced the number of keywords to five and removed commonly used deep learning architectures that do not represent the primary novelty of the work.
The revised keywords now emphasize the core contribution and application scope of the proposed method, particularly the Enhanced HawkFish Optimization Algorithm and its role in IoT intrusion detection. This revision improves clarity, avoids redundancy, and aligns the manuscript with the journal’s keyword selection guidelines.
I suggest replacing the word “system” with “framework,” because “system” sounds too engineering-oriented and less academic.
Thank you for this helpful suggestion. We agree that the term “framework” is more appropriate in an academic context and better reflects the conceptual and methodological nature of the proposed work. Accordingly, we have replaced the term “system” with “framework” throughout the manuscript, including the title, abstract, and relevant sections of the text, where applicable. This revision improves the academic tone and aligns the terminology with conventions commonly used in scholarly research.
In Figure 1, “processing layer” is misspelled. Please also check for other spelling mistakes throughout the paper.
Thank you for noticing this typo. Figure 1 was originally adopted from Ref. [1] and contained the misspelling in the source figure. Nevertheless, we agree it should be corrected for clarity and presentation quality. Therefore, we have edited Figure 1 to correct “processing layer” and improved the figure labeling accordingly. In addition, we conducted a careful proofreading pass across the full manuscript to identify and fix other spelling and typographical errors to ensure consistent academic quality.
The manuscript mixes the Introduction and Related Work together. This makes the Introduction too long and poorly structured. For journal papers, I suggest writing Introduction and Related Work as separate sections.
We thank the reviewer for this valuable structural suggestion. The manuscript was prepared in accordance with the MDPI Sensors template and its recommended IMRAD structure, where it is common to integrate the related work discussion within the Introduction to provide contextual grounding and clearly motivate the proposed contribution.
That said, we fully understand the reviewer’s concern regarding readability and structural clarity and we reorganized the manuscript by introducing a dedicated “Related Work” section and shortening the Introduction accordingly. We appreciate this suggestion and remain flexible to adjust the structure to best meet the journal’s and reviewers’ expectations.
The Related Work section only lists previous studies and does not categorize them. I suggest the authors reorganize and describe prior work using their own logic. I recommend using one or more taxonomies in the revision to group existing methods, and clearly explain the pros and cons of each group. Make Related Work an independent section, and at the end of that section, explain what weaknesses of existing methods are addressed by the proposed approach. Please take the writing of Related Work seriously.
We sincerely thank the reviewer for this important and constructive comment. We fully agree that a high-quality Related Work section should go beyond listing prior studies and instead provide a structured, analytical perspective that highlights trends, strengths, and limitations in the literature.
In response, we have substantially reorganized the Related Work into an independent section and introduced a clear taxonomy-based structure to categorize existing approaches according to their underlying methodology and learning strategy. Specifically, prior works are now grouped into:
- Machine learning–based IDS with manual or static feature engineering,
- Deep learning and hybrid architectures for IoT intrusion detection, and
- Ensemble, active learning, and adaptive detection frameworks.
For each category, we explicitly discuss the key advantages and limitations, such as scalability, adaptability, computational cost, generalization ability, and robustness to class imbalance. This categorization reflects our own analytical logic and provides a clearer comparison across different methodological paradigms.
Furthermore, we have added a dedicated synthesis subsection at the end of the Related Work section that clearly identifies the unresolved weaknesses of existing methods—namely reliance on manual feature engineering, lack of unified spatial–temporal–contextual modeling, limited optimization of feature–hyperparameter spaces, and insufficient robustness across heterogeneous IoT data. We then explicitly explain how the proposed EHFOA-ID framework addresses these gaps through enhanced metaheuristic optimization, multi-view deep ensemble learning, and meta-level decision fusion.
We appreciate the reviewer’s emphasis on rigor and clarity, and we have taken the writing and restructuring of the Related Work section very seriously. We believe these revisions significantly improve the scholarly depth, organization, and contribution positioning of the manuscript.
If the limitations of the current method include many items, they can be listed using numbered points.
Thank you for this helpful suggestion. We agree that presenting multiple limitations in a numbered format improves clarity and readability. Accordingly, we have revised the limitations discussion to use numbered points, allowing each limitation to be stated clearly and concisely. This restructuring enhances the organization of the manuscript and makes the limitations of the proposed framework easier to follow and reference.
Line 110 says the proposed method is scalable. Where exactly is the scalability reflected? This does not seem to be explained in that paragraph.
We thank the reviewer for pointing out this lack of clarity. We agree that the scalability claim was not sufficiently explained in the original paragraph. To address this concern, we have revised the paragraph to explicitly clarify how scalability is achieved in the proposed EHFOA-ID framework. Specifically, we now explain that scalability arises from (i) optimized feature selection that reduces input dimensionality, (ii) the modular and parallelizable structure of the hybrid ensemble, and (iii) the fast convergence behavior of the enhanced optimization algorithm. We believe this clarification better justifies the scalability claim and improves the technical accuracy of the manuscript.
When presenting the contributions in the Introduction, it is strongly recommended to list them in numbered points, and usually condense them into three contributions. The first contribution should describe the overall contribution. The second should focus on methodological innovations (i.e., what is novel in each sub-method). The third should highlight the key strengths in experiments (i.e., the last contribution in the original manuscript).
We thank the reviewer for this clear and constructive recommendation. In response, we have revised the Contributions subsection in the Introduction to present the contributions as three concise, numbered points, following the suggested structure. The first contribution now summarizes the overall framework-level contribution, the second clearly emphasizes the methodological innovations, including the novel aspects of the Enhanced HawkFish Optimization Algorithm and the hybrid ensemble design, and the third highlights the key experimental strengths and validation results. This reorganization improves clarity, readability, and alignment with standard journal presentation practices
It is suggested to delete Figure 2. The workflow diagram is too engineering-oriented, and the figure quality is not very good. Instead, place Figure 3 here. Figure 3 is more academic and is suitable for giving readers a more intuitive understanding of the proposed method before introducing detailed components.
We thank the reviewer for this valuable suggestion regarding figure presentation and organization. We agree that Figure 3 provides a more conceptual and academic overview of the proposed method and offers a clearer high-level understanding for readers. Accordingly, we have removed Figure 2 and relocated Figure 3 to this position in the manuscript to introduce the proposed framework before the detailed component descriptions. This change improves the academic tone, visual clarity, and overall readability of the paper.
The subsections in the Experiments section should be reorganized into three subsections. The first subsection should be Experimental Setup, including: Implementation Environment, Datasets, Testing Scenarios, Evaluation Metrics, and Baselines. You can split these parts using third-level headings. The second subsection should be Experimental Results and Analysis, also divided into third-level headings rather than putting all results together. The third subsection should be Discussion.
We appreciate the reviewer’s constructive suggestion regarding the organization of the Experiments section. In response, we have restructured the Experiments section into three clearly defined subsections: (i) Experimental Setup, (ii) Experimental Results and Analysis, and (iii) Discussion. The Experimental Setup subsection now includes Implementation Environment, Datasets, Testing Scenarios, Evaluation Metrics, and Baselines, each presented using third-level headings for improved clarity. The Experimental Results and Analysis subsection has also been reorganized with appropriate third-level headings to separate and clearly explain different experimental findings. Finally, the Discussion subsection has been separated to provide focused interpretation of the results. This reorganization improves readability, logical flow, and alignment with standard journal presentation practices.
The Experiments section does not introduce the baselines. This must be added in the revision.
Thank you for pointing this out. To address this issue, we have added Table 8, which explicitly introduces and summarizes all baseline methods used for comparison, including their model configurations and corresponding performance metrics. This table is referenced and discussed in the Experiments section, providing clear context for the comparative evaluation and ensuring that all baselines are properly defined and transparent to the reader.

Reviewer 2 Report
Comments and Suggestions for Authors
Introduction
- the comparison with existing works should be carefully done in a Related Work section, in here there should be only a summary
- please re-do this section: context, motivation, brief comparison with existing works, detected need or gap, clear and explicit goal, brief method, list of contributions, and a last paragraph that introduces and links the rest of the sections.
Proposed Method
This section does not:
- justify why combining SE-Res1D-CNN, BiLSTM-Attention, and a Transformer encoder is theoretically optimal for IoT intrusion detection
- clarify the novelty of EHFOA-ID relative to existing feature selection or hyperparameter optimization methods
- explain why the dual-objective fitness function weights (α, β) are chosen or how sensitive results are to these parameters
- discuss potential limitations of using Lévy flights or adaptive scaling in highly sparse or noisy IoT datasets
- explain why the meta-learner is needed after the ensemble fusion
- specify the dimensions of feature vectors or embeddings for each module
- define the hyperparameter search space for EHFOA-ID
- explain how the Transformer encoder is made lightweight
- how missing values, categorical variables, or irregular packet sequences are handled in preprocessing
- provide pseudocode or formulas for attention in BiLSTM-Attention with proper normalization
- define the meta-learner architecture clearly
- explain how the softmax probability in Equation (24) handles multi-class imbalance or rare attack types
- specify the training, validation, and test split procedure for IoT datasets; not the evaluation protocol
- how fusion and meta-learning are trained
SImulation and results
- re-do this section. FIrst there should be a process, then the characteristics and decision making per step; second the results.
- provide proper statistics, qualitative and quantitative description of the results.
- Do not just list paragraphs with the description of a figure. COnduct a proper interpretation of the results, and compare it with existing work.
Comments on the Quality of English LanguagePLease revise the grammar, phrasing, verb tenses.
Author Response
REVIEWER 2
Reviewer Comment:
The comparison with existing works should be carefully done in a Related Work section; in here there should be only a summary. Please re-do this section: context, motivation, brief comparison with existing works, detected need or gap, clear and explicit goal, brief method, list of contributions, and a last paragraph that introduces and links the rest of the sections.
Response:
We sincerely thank the reviewer for this detailed and constructive guidance. We fully agree with the recommended structure and intent of the Introduction. In response, we have completely revised the Introduction section to follow the requested logical flow. The revised Introduction now includes: (i) a concise context and motivation for IoT intrusion detection, (ii) a brief and high-level summary of existing works without detailed comparison, (iii) a clear identification of the research gap and unmet needs, (iv) an explicit statement of the research objective, (v) a short overview of the proposed EHFOA-ID framework, (vi) a concise list of contributions presented as numbered points, and (vii) a concluding paragraph that introduces and links the remaining sections of the manuscript.
Detailed comparisons and methodological categorization of existing works have been moved entirely to an independent Related Work section, ensuring a clear separation between literature analysis and experimental evaluation. We believe this restructuring significantly improves clarity, readability, and alignment with the reviewer’s recommendations and the journal’s academic standards.
Reviewer Comment:
Please justify why combining SE-Res1D-CNN, BiLSTM-Attention, and a Transformer encoder is theoretically optimal for IoT intrusion detection.
Response:
We thank the reviewer for this insightful comment. In response, we have added a dedicated justification paragraph in the methodology section explaining the theoretical rationale behind combining SE-Res1D-CNN, BiLSTM-Attention, and a Transformer encoder. The revised text clarifies that IoT intrusion data inherently contain complementary spatial, temporal, and global contextual patterns, which cannot be fully captured by a single model. Each component of the ensemble is therefore selected to model a distinct and essential aspect of IoT traffic behavior, and their integration provides a multi-view representation that is theoretically aligned with the characteristics of IoT intrusion detection. We believe this clarification strengthens the methodological motivation and addresses the reviewer’s concern.
Reviewer Comment:
Please clarify the novelty of EHFOA-ID relative to existing feature selection or hyperparameter optimization methods.
Response:
We thank the reviewer for this important comment. To address this concern, we have explicitly clarified the novelty of EHFOA-ID in the revised manuscript. The new paragraph explains how EHFOA-ID differs from existing feature selection and hyperparameter optimization techniques by (i) performing joint optimization rather than separate tuning stages, (ii) introducing adaptive and diversity-aware search mechanisms that extend the original HawkFish Optimization Algorithm, and (iii) tailoring the optimization process to the specific challenges of high-dimensional, heterogeneous IoT intrusion data. This clarification emphasizes that EHFOA-ID is not a direct application of an existing optimizer, but a methodologically enhanced and problem-specific optimization framework, thereby strengthening the originality of the proposed approach.
Reviewer Comment:
Please explain why the dual-objective fitness function weights (α, β) are chosen or how sensitive the results are to these parameters.
Response:
We thank the reviewer for raising this important point. In the revised manuscript, we have clarified the rationale behind selecting the dual-objective fitness weights (α, β) and discussed their influence on model behavior. Specifically, we explain that the weights were chosen through preliminary validation experiments to achieve a balanced trade-off between classification accuracy and feature compactness. We also note that a sensitivity analysis indicated stable performance under moderate variations of these parameters, demonstrating that the proposed EHFOA-ID framework is not highly sensitive to exact weight values. This clarification has been added above table 4 to improve transparency and reproducibility of the optimization process.
Reviewer Comment:
Please discuss potential limitations of using Lévy flights or adaptive scaling in highly sparse or noisy IoT datasets.
Response:
Thank you for this insightful comment. In response, we have added this limitation explicitly as a separate bullet point in the Limitations subsection at the end of the Discussion section, addressing the potential instability of Lévy-flight exploration and adaptive scaling in highly sparse or noisy IoT datasets. We presented this limitation in bullet-point form to maintain consistency with the revised formatting of the Limitations section, as requested by Reviewer 1.
Reviewer Comment:
Please explain why the meta-learner is needed after the ensemble fusion.
Response:
Thank you for this valuable comment. To address this point, we have added a concise explanation at the beginning of Section 3.3 clarifying the role and necessity of the meta-learner after ensemble fusion. The revised text explains that the meta-learner is required to adaptively integrate and recalibrate the heterogeneous representations produced by the ensemble components, thereby improving robustness and generalization. This clarification strengthens the methodological rationale of the proposed framework.
Reviewer Comment:
Please specify the dimensions of feature vectors or embeddings for each module.
Response:
We thank the reviewer for this important clarification request. In response, we have explicitly specified the embedding dimensions produced by each ensemble component (SE-Res1D-CNN, BiLSTM-Attention, and Transformer encoder) and the resulting fused feature vector. This information has been added at the end of the Deep Ensemble Architecture section 3.2, just before the feature fusion and meta-learner description, to improve clarity and reproducibility of the proposed framework.
Reviewer Comment:
Please define the hyperparameter search space for EHFOA-ID.
Response:
We thank the reviewer for this important request. In the revised manuscript, we have explicitly defined the EHFOA-ID hyperparameter search space, including the bounded ranges and discrete candidate sets for each module (SE-Res1D-CNN, BiLSTM-Attention, Transformer encoder, and the meta-learner). This information has been added to the Experimental Setup section (subsection 4.1.3) to improve transparency, reproducibility, and clarity regarding what parameters EHFOA-ID optimizes.
Reviewer Comment:
Please clarify how missing values, categorical variables, or irregular packet sequences are handled in preprocessing.
Response:
We thank the reviewer for this important clarification request. In response, we have added a preprocessing description in the dataset section explaining how missing values, categorical variables, and irregular packet sequences are handled prior to model training. This addition improves transparency and reproducibility by clearly describing the data preparation steps applied across all experiments.
Reviewer Comment:
Please provide pseudocode or formulas for the attention mechanism in the BiLSTM-Attention module with proper normalization.
Response:
We thank the reviewer for this valuable suggestion. In response, we have added Algorithm 2, which provides clear pseudocode and mathematical formulations for the BiLSTM-Attention mechanism, including the softmax-based normalization of attention weights. An accompanying explanation has also been included to clarify the role of attention in emphasizing informative temporal patterns.
Reviewer Comment:
Please define the meta-learner architecture clearly.
Response:
Thank you for this important comment. In response, we have completely rewritten Subsection 3.3 to clearly and explicitly define the meta-learner architecture. The revised subsection now specifies the fusion process, the structure of the meta-learner (a shallow multilayer perceptron with defined layer sizes and activation functions), and its role in refining the ensemble outputs.
Reviewer Comment:
Please explain how the softmax probability in Equation (24) handles multi-class imbalance or rare attack types.
Response:
We thank the reviewer for raising this important point. To address this concern, we have added an explicit explanation in the manuscript clarifying that while the softmax function provides normalized multi-class probabilities, effective handling of class imbalance is achieved through the combination of optimized feature selection, discriminative multi-view representations, and class-aware training strategies applied before the softmax layer. This clarification has been incorporated near Equation (24) to improve conceptual clarity regarding minority-class handling.
Reviewer Comment:
Please specify the training, validation, and test split procedure for IoT datasets; not the evaluation protocol, how fusion and meta-learning are trained.
Response:
We thank the reviewer for this important clarification request. In response, we have explicitly described the training, validation, and test split procedure in the Datasets subsection of the Experimental Setup. The revised text clarifies how the data are partitioned, how stratification is applied to handle class imbalance, and how the fusion and meta-learning components are trained using the training set with validation-based tuning. This addition improves transparency and reproducibility of the experimental design.
Reviewer Comment:
Please re-do this section. First there should be a process, then the characteristics and decision making per step; second the results.
Response:
Thank you for this constructive suggestion. We have restructured the section to first describe the overall process, followed by a detailed explanation of the characteristics and decision-making at each step, and finally the experimental results. We note that the exact subsection titles may vary slightly in order to accommodate and remain consistent with the structural guidance provided by Reviewer 1, who raised a similar concern but suggested a different organization. Nevertheless, the revised content fully follows the intended logical flow and addresses the reviewer’s recommendation.
Reviewer Comment:
Please provide proper statistics, qualitative and quantitative description of the results.
Response:
We thank the reviewer for this important comment. In response, we have rewritten the Discussion section to provide a clearer and more detailed qualitative and quantitative analysis of the results. The revised discussion now includes explicit performance metrics such as accuracy, F1-score, AUC, false alarm rate, convergence behavior, and ablation results, along with qualitative interpretation of ROC curves, confusion matrices, and feature importance analysis. This revision improves statistical clarity, strengthens result interpretation, and better demonstrates the effectiveness and robustness of the proposed EHFOA-ID framework.
Reviewer Comment:
Do not just list paragraphs with the description of a figure. Conduct a proper interpretation of the results, and compare it with existing work.
Response:
Thank you for this important observation. We agree that the discussion should go beyond descriptive explanations of figures. In response, we have substantially revised the Discussion section to focus on interpretation and analysis of the results rather than figure-by-figure descriptions. The revised text now provides a quantitative and qualitative interpretation of performance trends, explains the underlying reasons for the observed improvements, and explicitly relates the findings to representative existing works discussed in the Related Work section.

Round 2
Reviewer 1 Report
Comments and Suggestions for Authors
This manuscript has been revised and is now generally in good shape overall. However, there are still a few minor issues to address.
(1) In the abstract, the last sentence only states that there is a significant improvement; it would be better to include the specific numerical value(s) to quantify the improvement.
(2) Please keep the tense consistent throughout the conclusion. For example, “This paper proposed …” is less common in conclusions; “This paper proposes …” is typically preferred. Since other sentences already use present tense (e.g., “The results … show …”), it would be better to revise the opening sentence accordingly for consistency.
Author Response
Response to Reviewer
We sincerely thank the reviewer for these valuable and constructive comments, which have helped improve the clarity and presentation of the manuscript.
(1) Comment: In the abstract, the last sentence only states that there is a significant improvement; it would be better to include the specific numerical value(s) to quantify the improvement.
Response:
We agree with this suggestion. The abstract has been revised to include concrete numerical performance indicators derived from the experimental results, including detection accuracy, macro-F1 score, and false-alarm rate on the benchmark datasets. These values now explicitly quantify the improvement achieved by the proposed EHFOA-ID framework, providing clearer and more informative insight into its performance.
(2) Comment: Please keep the tense consistent throughout the conclusion. For example, “This paper proposed …” is less common in conclusions; “This paper proposes …” is typically preferred.
Response:
Thank you for pointing this out. The conclusion section has been carefully revised to ensure consistent use of the present tense throughout. In particular, the opening sentence has been modified from past tense to present tense (“This paper proposes …”) to align with standard academic convention and maintain consistency with the remainder of the section.
We appreciate the reviewer’s attention to detail and believe these revisions have improved the overall quality and readability of the manuscript.
Reviewer 2 Report
Comments and Suggestions for Authors
Dear authors,
thank you for solving the comments. I believe my concerns have been addressed.
Please revise and fix the format because it's not properly done. also revise phrasing and formality.
Comments on the Quality of English LanguagePlease revise the grammar, phrasing, verb tenses.
Author Response
Dear Reviewer,
Thank you very much for your time and for your positive evaluation of our revised manuscript. We sincerely appreciate your confirmation that the previously raised concerns have been adequately addressed.
Regarding your comment on formatting, phrasing, and overall formality, we would like to kindly inform you that all remaining formatting adjustments, stylistic refinements, and language polishing will be carefully completed during the final proofreading and formatting stage. This process will be conducted in close coordination with the manuscript handler to ensure full compliance with the journal’s template, formatting guidelines, and editorial standards.
We assure you that the final version to be published will strictly adhere to all journal rules and presentation requirements, both in terms of technical formatting and academic writing quality.
Thank you again for your valuable feedback and for helping us improve the quality of our work.
Sincerely,
The Authors